# CDK-mediated phosphorylation of PNKP is required for end-processing of single-strand DNA gaps on Okazaki fragments and genome stability

**Kaima Tsukada[1,2]\*[†], Rikiya Imamura[1][†], Tomoko Miyake[1], Kotaro Saikawa[1], Mizuki Saito[1], Naoya Kase[1], Lingyan Fu[1], Masamichi Ishiai[3], Yoshihisa Matsumoto[1], Mikio Shimada[1]\***

[1]Laboratory for Zero-Carbon Energy, Institute of Integrated Research, Institute of Science Tokyo, Tokyo, Japan; [2]Center for Chromosome Stability, Department of Cellular and Molecular Medicine, University of Copenhagen, Copenhagen, Denmark; [3]National Cancer Center Research Institute, Tokyo, Japan

**Abstract** Polynucleotide kinase phosphatase (PNKP) has enzymatic activities as 3′-phosphatase and 5′-kinase of DNA ends to promote DNA ligation and repair. Here, we show that cyclin-dependent kinases (CDKs) regulate the phosphorylation of threonine 118 (T118) in PNKP. This phosphorylation allows recruitment to the gapped DNA structure found in single-strand DNA (ssDNA) nicks and/or gaps between Okazaki fragments (OFs) during DNA replication. T118A (alanine)-substituted PNKP-expressing cells exhibited an accumulation of ssDNA gaps in S phase and accelerated replication fork progression. Furthermore, PNKP is involved in poly (ADP-ribose) polymerase 1 (PARP1)-dependent replication gap filling as part of a backup pathway in the absence of OFs ligation. Altogether, our data suggest that CDK-mediated PNKP phosphorylation at T118 is important for its recruitment to ssDNA gaps to proceed with OFs ligation and its backup repairs via the gap-filling pathway to maintain genome stability.

**\*For correspondence:**
kaimat@sund.ku.dk (KT);
mshimada@zc.iir.titech.ac.jp (MS)

[†]These authors contributed equally to this work

**Competing interest:** The authors declare that no competing interests exist.

## Editor's evaluation

PNKP (polynucleotide kinase phosphatase) has dual functions of adding or removing phosphate groups at broken DNA ends for DNA damage repair, such as base damage or strand breaks. Here the authors provide convincing evidence that PNKP is also involved in the DNA replication process, particularly in processing Okazaki fragment ends. This is an important finding that expands our understanding of PNKP function, showing that it extends beyond DNA damage repair.

## Introduction

Genomic DNA is threatened by intrinsic factors such as oxidative stress derived from mitochondria-dependent energy metabolism and extrinsic factors such as ionizing radiation (IR), ultraviolet radiation, and chemical compounds. These factors generate various types of DNA damage, such as base damage, DNA single-strand breaks (SSBs), and double-strand breaks (DSBs). Because DNA lesions cause chromosomal aberrations, leading to aneuploidy and tumorigenesis, DNA repair machinery has evolved in living organisms. Polynucleotide kinase phosphatase (PNKP) is a key enzyme with a dual role that bears 3′-phosphatase and 5′-kinase activity (*Chappell et al., 2002*; *Karimi-Busheri et al., 1999*; *Jilani et al., 1999*; *Coquelle et al., 2011*). Human PNKP consists of 521 amino acids, including

the forkhead-associated (FHA) domain (amino acid residues 1–110) in the amino-terminal region, and phosphatase (146–337) and kinase (341–516) domains in the carboxy-terminal region, which are connected by a linker region (111–145). PNKP is recruited to SSBs and DSBs sites depending on its interactions via the FHA domain of PNKP with XRCC1 and XRCC4, respectively, and is involved in base excision repair (BER), SSB repair, and non-homologous end joining for DSB repair (*Breslin and Caldecott, 2009*; *Mani et al., 2019*; *Mani et al., 2010*; *Tsukada et al., 2020*; *Tsukada et al., 2021*). The PNKP linker region includes a nuclear localization signal and phosphorylation sites (*Tsukada et al., 2020*). Phosphorylation of PNKP at serine 114 by ataxia telangiectasia mutated (ATM) is required for protein stability and efficient DNA repair for cellular survival (*Segal-Raz et al., 2011*; *Parsons et al., 2012*). Mutations in PNKP are associated with the human inherited disease microcephaly and seizures (MCSZ), a neurodevelopmental disease (*Shen et al., 2010*), ataxia oculomotor apraxia 4 (AOA4) (*Bras et al., 2015*), and Charcot–Marie–Tooth disease (CMT2B2), a neurodegenerative disease (*Pedroso et al., 2015*). These mutations are mostly located in the phosphatase or kinase domains and attenuate the phosphatase and kinase activities (*Reynolds et al., 2012*; *Kalasova et al., 2020*; *Bermúdez-Guzmán et al., 2020*).

The BER and SSBs repair intermediates form gapped DNA structures, which are the main targets of PNKP. These DNA gaps are also found in Okazaki fragments (OFs) during DNA replication. DNA replication integrity is essential for ensuring genomic stability and accurate cell proliferation (*Siddiqui et al., 2013*; *O'Donnell et al., 2013*). DNA replication is initiated at the origin of replication in the leading strand and at short RNA-primed DNA fragments, known as OFs, in the lagging strand (*Leonard and Méchali, 2013*; *Okazaki et al., 1968*). Recent reports suggest that single-strand DNA (ssDNA) gap-filling machinery is involved in OFs maturation (*Hanzlikova et al., 2018*). Poly (ADP-ribose) polymerase 1/2 (PARP1/2) is involved in the repair of SSBs, DSBs, and multiple DNA replication processes (*Ray Chaudhuri et al., 2012*; *Ray Chaudhuri and Nussenzweig, 2017*). PARP1 activity is required for efficient SSBs repair and ssDNA gap-filling pathway in OFs during DNA replication (*Hanzlikova et al., 2018*). The DNA replication machinery is precisely controlled by cyclin-dependent kinases (CDKs), which phosphorylate several replication factors to allow them to enter the S phase and promote DNA synthesis (*Swaffer et al., 2016*). During replication fork progression, ssDNA is fragile and protected by replication protein A (RPA) (*Wold, 1997*). When replication forks stall, ataxia telangiectasia mutated and Rad3-related protein (ATR) is activated and phosphorylates CHK1 and RPA to resolve or eliminate the impediment, promoting replication fork recovery (*Cortez et al., 2001*; *Zou and Elledge, 2003*). Furthermore, DNA damage, such as base damage, SSBs, and DSBs, occurs upon genotoxic stress and replication errors, which lead to DNA replication stress, genomic instability, genetic mutation, and tumorigenesis.

In this study, we found that PNKP was required for the ssDNA gap-filling pathway during DNA replication. Defects in PNKP induce the accumulation of single-strand gaps in OFs and genome instability. We also found that CDKs phosphorylate PNKP on threonine 118 (T118), mainly in the S phase, and that CDK-mediated PNKP phosphorylation allows it to be recruited to ssDNA gaps on OFs. Moreover, PNKP enzymatic activities, especially phosphatase activity, are required for processing of the ends of single-strand gaps on OFs. Taken together, our data suggest that phosphorylation-mediated PNKP recruitment to ssDNA gaps on OFs and the end-processing activity of PNKP are critically important for preventing genome instability through appropriate regulation of DNA replication.

## Results
### Generation of PNKP knockout U2OS cell line
In a previous study, we found that the depletion of PNKP resulted in impaired cell growth in mice (*Shimada et al., 2015*). To confirm that this phenotype is observed in human cells, we initially generated PNKP knockout U2OS cells (*PNKP*⁻/⁻ cells) using CRISPR/Cas9 Nickase (D10A) targeting exon 4 of the *PNKP* coding region (*Chiang et al., 2016*). We obtained two clones (C1 and C2) that showed complete loss of PNKP, as confirmed by western blot analysis with both N-terminus and C-terminus PNKP-recognized antibodies and DNA sequencing (*Figure 1*, *Figure 1—figure supplement 1A, B*).

To confirm that these *PNKP*⁻/⁻ cells have functional deficiencies in DNA repair, we analyzed their DSB repair ability (*Figure 1—figure supplement 2A, B*). *PNKP*⁻/⁻ cells showed a delay in diminishing the phosphorylation of histone H2AX and KAP1, markers of DSBs, after IR-induced DNA damage

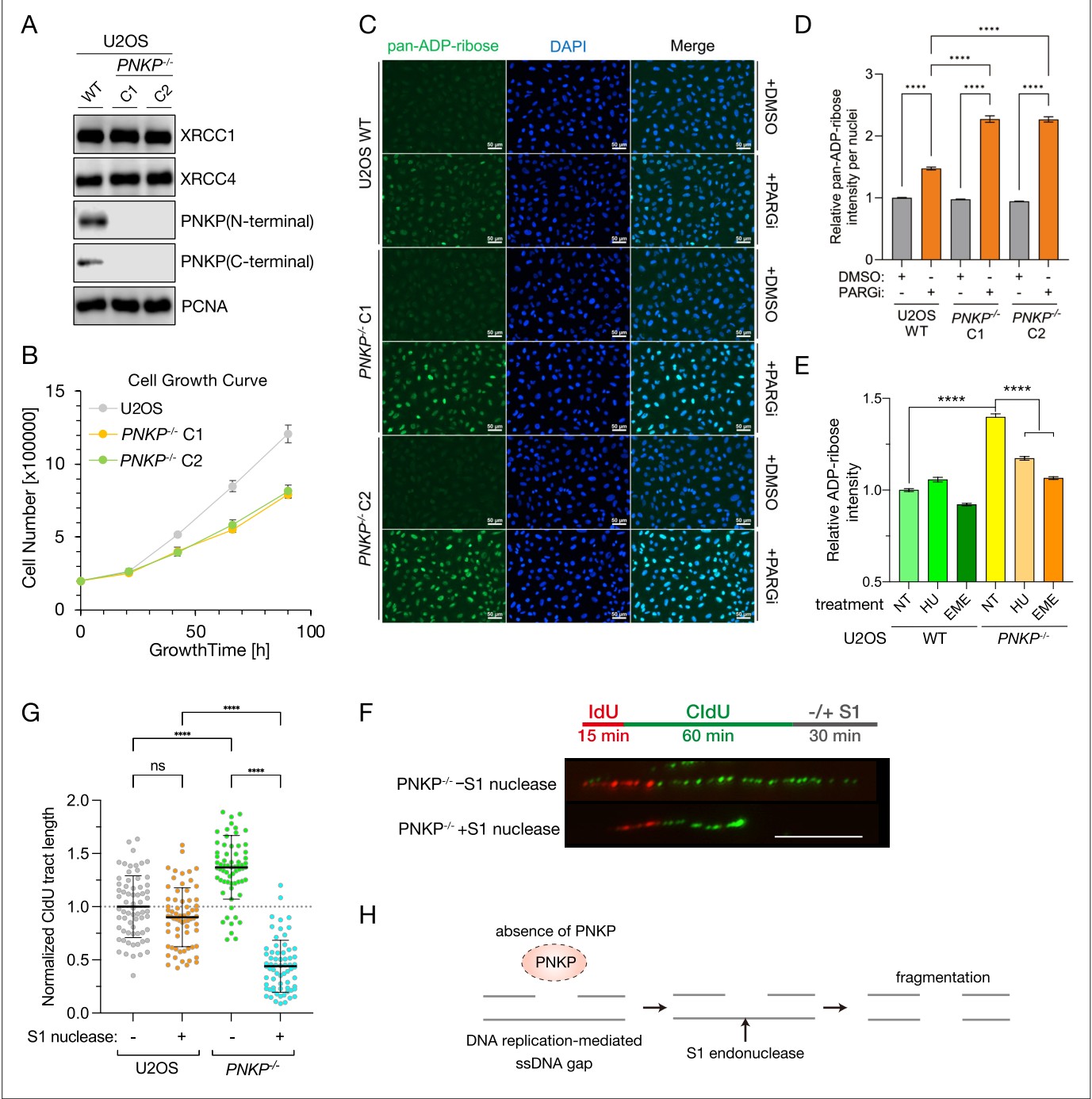

**Figure 1.** Loss of polynucleotide kinase phosphatase (PNKP) causes delayed cell proliferation due to accumulated single-strand DNA gaps in S phase. (**A**) Protein expression analysis of PNKP knockout U2OS cells (*PNKP*−/− cells) generated by CRISPR/Cas9 genome editing. Protein expression of PNKP in *PNKP*−/− clone 1 (C1) and clone 2 (C2) were confirmed by western blotting with N- and C-terminal recognized PNKP antibodies, respectively. XRCC1, XRCC4, and PCNA antibodies were used as loading controls. (**B**) Measurement of growth rate of U2OS WT and of *PNKP*−/− C1 and C2 cells. Cell numbers (shown in vertical axis) were counted at indicated time points (shown in horizontal axis). (**C, D**) Measurement of endogenous DNA single-strand breaks (SSBs) of *PNKP*−/− cells. SSBs were analyzed by immunofluorescence using PAN-ADP-ribose-binding reagents at 30 min after ionizing-radiation (IR) 2 Gy exposure in U2OS WT and *PNKP*−/− (C1 and C2) cells treated with 10 µM poly ADP-ribose glycohydrolase inhibitor (PARGi) for 60 min. Error bars represent standard error of the mean (SEM). (**E**) Measurement of SSBs, detected by ADP-ribose intensity, in U2OS WT and *PNKP*−/− C1 cells under hydroxyurea (HU) or Emetine (EME) treatment. Cells were pretreated with PARGi for 30 min prior to the treatment of HU or EME. ADP-ribose intensities were normalized by the intensity of non-treated (NT) U2OS. Error bars represent SEM. (**F**) Schematic and a representative image of experiments for measuring the formation of single-strand DNA gaps during DNA replication in U2OS WT and *PNKP*−/− C1 cells with S1 nuclease treatment.

*Figure 1 continued on next page*

*Figure 1 continued*

(**G**) Quantified results of DNA fiber length in U2OS WT and *PNKP*⁻/⁻cells with S1 nuclease. CldU tract lengths in CldU/IdU dual-labeled DNA fibers in the indicated cell lines were plotted as scatter plots. Error bars represent standard deviation (SD). Statistical significance was indicated as not significant (ns) and ****: 0.0005 < p ≦ 0.001. (**H**) Model of S1 nuclease-mediated digestion of DNA fiber in *PNKP*⁻/⁻ cells. In all panels, scale bar indicates 10 µm.

The online version of this article includes the following source data and figure supplement(s) for figure 1:

**Source data 1.** Original membranes corresponding to *Figure 1A*.

**Source data 2.** Original membranes corresponding to *Figure 1A*.

**Figure supplement 1.** Generation of polynucleotide kinase phosphatase (PNKP) knockout U2OS cells by genome editing.

**Figure supplement 2.** Polynucleotide kinase phosphatase (PNKP)-deficient cells exhibit genomic instability to various genotoxic stresses.

**Figure supplement 2—source data 1.** Original membranes corresponding to *Figure 1—figure supplement 2A*.

**Figure supplement 2—source data 2.** Original membranes corresponding to *Figure 1—figure supplement 2A*.

(*Figure 1—figure supplement 2A*). *PNKP*⁻/⁻ cells also showed increased sensitivity to IR exposure, suggesting a defect in DSB repair ability (*Figure 1—figure supplement 2B*). We then assessed SSB repair activity in *PNKP*⁻/⁻ cells using a BrdU incorporation assay with in vitro Exonuclease III (ExoIII) digestion (*Figure 1—figure supplement 2C–E*). ExoIII excises nucleotides from 3′ ends of nick and/or gap associated with SSBs, resulting in exposing incorporated BrdU (*Figure 1—figure supplement 2C*). The cells displayed a significantly increased level of native BrdU signals and hyper-sensitivity upon SSB induction by hydrogen peroxide (H₂O₂), indicating that the loss of PNKP leads to reduced ability of SSB repair in cells (*Figure 1—figure supplement 2C–F*). These findings confirm deficiencies in both SSB and DSB repair in *PNKP*⁻/⁻ cells, aligning with observations from previous studies (*Chalasani et al., 2018*). Furthermore, *PNKP*⁻/⁻ cells showed increased sensitivity to the inhibition of DNA replication by hydroxyurea (HU) treatment (*Figure 1—figure supplement 2G*). Taken together, these results indicate that *PNKP*⁻/⁻ cells were successfully established.

## Loss of PNKP causes delayed cell proliferation due to accumulated ssDNA gaps in the S phase

During the course of our experiments, we observed that *PNKP*⁻/⁻ cells grow slower than WT cells, consistent with our previous study in mice (*Figure 1B*; *Shimada et al., 2015*). Since cell proliferation is associated with DNA replication, we analyzed the cell cycle distribution of *PNKP*⁻/⁻ cells using flow-cytometry. The analysis revealed an accumulation of *PNKP*⁻/⁻ cells in the S phase (*Figure 1—figure supplement 2H*). Considering that increased endogenous genotoxic stress can lead to improper S phase progression, we assessed the endogenous level of DNA damage in our two *PNKP*⁻/⁻ clones. Given that one of the known targets of PNKP in DNA repair is SSBs, including ssDNA gap structures, we anticipated that PNKP might be involved in the formation of ssDNA gaps during cell proliferation, not only in the repair of exogenous DNA damage. To test this, we performed PAN ADP-ribosylation assays combined with poly ADP-ribose glycohydrolase inhibitor (PARGi) treatment (*Figure 1C, D*; *Tsukada et al., 2020*; *Kalasova et al., 2020*). Our two *PNKP*⁻/⁻ clones exhibited significantly higher levels of ADP-ribosylation, a marker of ssDNA gaps/breaks, compared to WT cells, indicating that *PNKP*⁻/⁻ cells face a pronounced amount of endogenously induced ssDNA gaps/breaks. To investigate whether the accumulated ssDNA gaps are generated in S phase during DNA replication, we assessed PAN ADP-ribose levels in *PNKP*⁻/⁻ cells in combination with treatment with HU or Emetine (EME), a DNA replication inhibitor that blocks single-strand gap formation on replication forks via proteosynthesis inhibition (*Figure 1E*; *Burhans et al., 1991*; *Lukac et al., 2022*). In WT cells, HU treatment increased the amount of poly ADP-ribosylation, yet statistically non-significant (p = 0.067), whereas EME prevented poly ADP-ribosylation by inhibiting single-strand nick and/or gap formation in the S phase of U2OS WT cells (p = 0.0093). These results were consistent with previous studies (*Hanzlikova et al., 2018*). Although *PNKP*⁻/⁻ cells showed high levels of ADP-ribose intensity spontaneously, HU and EME treatment rescued the increased ADP-ribose intensity in *PNKP*⁻/⁻ cells, albeit higher, yet statistically non-significant, extent than WT cells. This suggests that loss of PNKP leads to an accumulation of ssDNA gap structures during DNA replication. To directly detect DNA replication-mediated ssDNA gap formation in *PNKP*⁻/⁻ cells, we performed an S1 DNA fiber assay (*Figure 1F, G*). Cells were labeled with 5-iodo-2'-deoxyuridine (IdU) for 15 min, followed by 60 min of 5-chloro-2'-deoxyuridine (CldU) labeling. The individual tract lengths of CldU-labeled nascent DNA in IdU/CldU-double positive

DNA fiber were quantified. Where indicated, labeled DNA was treated with a single-stranded DNA/RNA-specific endonuclease, S1 nuclease, to detect post-replicative ssDNA gaps by the shortened DNA tract length due to digestion of ssDNA gaps by S1 nuclease (*Figure 1H*). Despite the slower cell growth, *PNKP*$^{-/-}$ cells exhibited faster progression of DNA replication fork than WT cells (*Figure 1G*). This result suggests that PNKP may be involved in the progression of DNA replication forks, as a similar phenotype has been reported in PCNA polyubiquitination mutant (K164R) cells, which were unable to reduce the replication fork speed (*Thakar et al., 2020*). PCNA KR mutant cells also showed slow cell proliferation due to accelerated replication speed failing to protect nascent DNA from degradation. This resulted in a slower cell proliferation phenotype, reminiscent of that observed in *PNKP*$^{-/-}$ cells, although indeed recent study revealed that PNKP is involved in the fork protection (*Mashayekhi et al., 2024*). In addition, *PNKP*$^{-/-}$ cells exhibited significantly shortened DNA fibers upon S1 nuclease-mediated DNA digestion, unlike U2OS WT cells. This observation suggests that the loss of PNKP results in the accumulation of ssDNA gaps on nascent replicated DNA, leading to improper cell proliferation due to high levels of endogenous DNA damage (*Figure 1H*).

## PNKP is involved in canonical OFs maturation and ssDNA gap-filling pathways

The phenotype of increased tract length in *PNKP*$^{-/-}$ cells is observed in two of *PNKP*$^{-/-}$ clones, indicating that this phenotype is not clonal issue (*Figure 2A*). This phenotype in *PNKP*$^{-/-}$ cells is evocative of deficiencies in DNA replication factors involved in canonical OFs maturation, such as FEN1 and LIG1, or factors in the ssDNA gap-filling pathway, an alternative/backup pathway of OFs maturation, such as PARP1 (*Thakar et al., 2020*; *Maya-Mendoza et al., 2018*; *Cong et al., 2021*). Considering these points, we hypothesized that PNKP plays a critical role in preventing the accumulation of ssDNA gaps on replicating DNA during OFs maturation. To test this hypothesis, we first assessed DNA tract length in cells treated with HU or a flap endonuclease 1 inhibitor (FEN1i), which prevents the resection of overhanging nucleotides from the ends of OFs, leading to unligated OFs (*Figure 2B*, *Figure 2—figure supplement 1*; *Exell et al., 2016*; *Ward et al., 2017*; *Zheng and Shen, 2011*). Indeed, FEN1i treatment induced increased CldU tract length, whereas HU treatment led to stalled forks, consistent with previous studies (*Thakar et al., 2020*). Upon deficient canonical FEN1-mediated OFs maturation, post-replicative ssDNA gaps arise from unligated OFs, which are sensed by PARP1 and repaired via the PARP1-dependent gap-filling pathway for OFs maturation (*Vaitsiankova et al., 2022*). The PARP1-dependent gap-filling pathway is associated with XRCC1, a binding scaffold protein of PNKP in the SSB repair pathway (*Hanzlikova et al., 2018*). To determine whether PNKP plays a critical role in the PARP-dependent gap-filling pathway, we performed an S1 DNA fiber assay (*Figure 2C, D*). Cells were labeled with IdU for 15 min, followed by 60 min of CldU labeling in the presence or absence of an FEN1 inhibitor and/or PARP inhibitor, and the ratio of the individual tract lengths of nascent DNA labeled with IdU or CldU were quantified. Where indicated, labeled DNA was treated with S1 nuclease. Double treatment of FEN1i and PARPi in U2OS WT cells, followed by S1 nuclease treatment, showed significantly lower CldU/IdU ratio, indicating extensive DNA fiber digestion by S1 nuclease (*Figure 2C*). Although single treatment with either FEN1i or PARPi in U2OS WT cells, followed by S1 nuclease treatment, led to DNA fiber digestion, the extent was limited compared to the double treatment (*Figure 2C, D*), indicating that both pathways: FEN1-dependent canonical OFs maturation and PARP-dependent ssDNA gap-filling pathway, are coordinately required to prevent the emergence of ssDNA gaps during DNA replication, consistent with previous studies (*Vaitsiankova et al., 2022*). On the other hand, *PNKP*$^{-/-}$ cells with S1 nuclease treatment showed extensive DNA fiber digestion even without FEN1i and PARP1i treatments, and this was not further increased by FEN1i and PARPi treatment. These results suggest that PNKP itself is involved in both pathways mentioned above (*Figure 2E*). Therefore, loss of PNKP results in a phenotype similar to the loss of FEN1 in terms of canonical OFs maturation, but also, like PARP inhibition, there is an additional effect in repairing ssDNA gaps created under FEN1 loss conditions.

## PNKP phosphorylation, especially of T118, is important for proper S phase progression and cell proliferation

To identify the region of PNKP involved in proper fork progression via the OFs maturation processes, we measured the growth rate and speed of fork progression in *PNKP*$^{-/-}$ cells transiently expressing

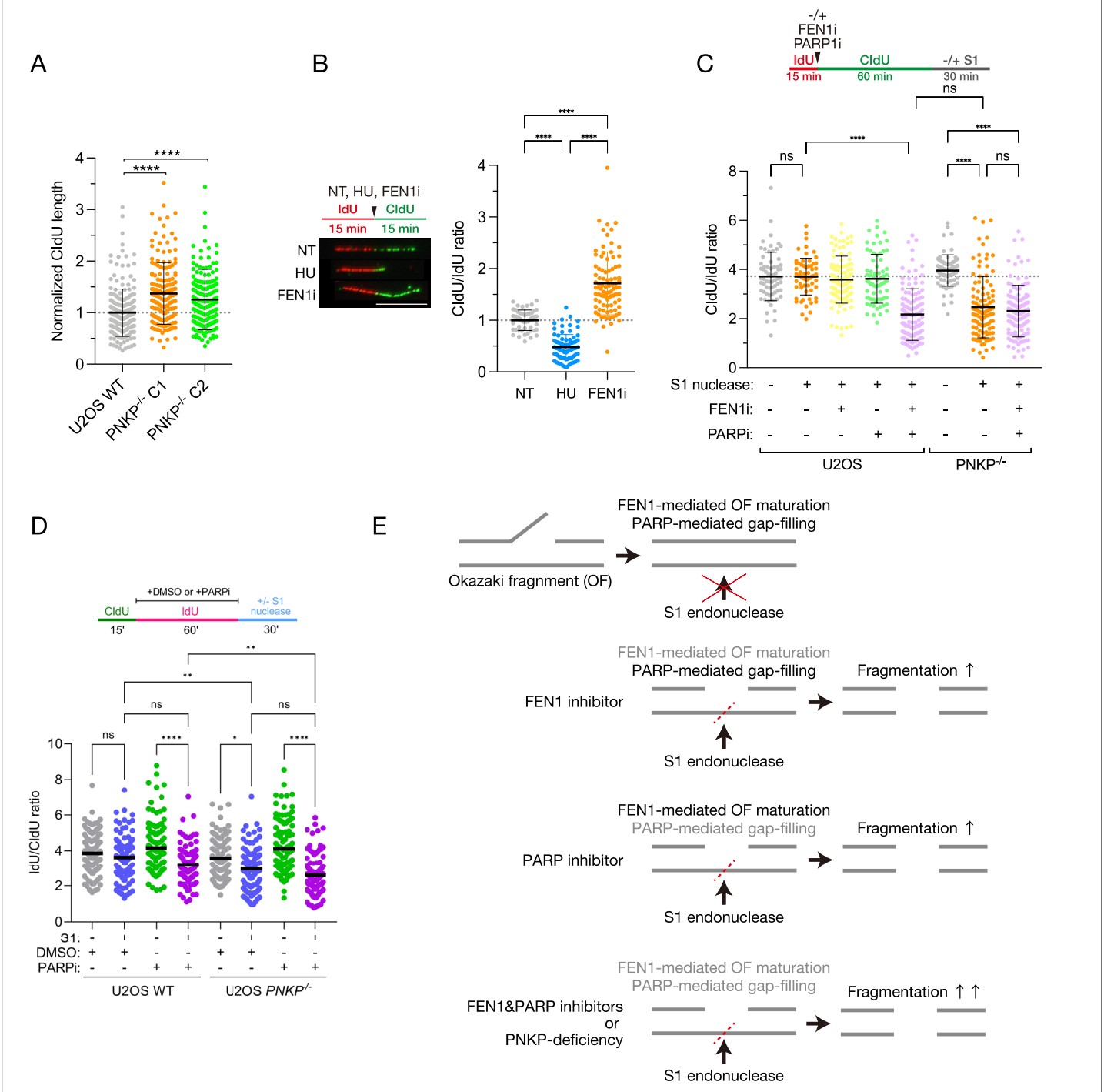

**Figure 2.** Polynucleotide kinase phosphatase (PNKP) is involved in Okazaki fragment maturation pathways. (**A**) Quantified results of DNA fiber length in U2OS WT and *PNKP*$^{-/-}$ C1 and C2 cells. Nascent synthesized DNA was labeled by CldU. CldU tract lengths in the indicated cell lines were plotted as scatter plots of fork speed. At least 100 fibers per sample were quantified. Error bars represent SD. (**B**) Measurement of DNA synthesis speed in U2OS WT cells under NT, 2 mM hydroxyurea (HU) and 10 μM FEN1 inhibitor (FEN1i) treatment analyzed by DNA fiber assay. At least 100 DNA fibers were measured. Error bars represent SD. (**C**) Schematic and quantified results of experiments for measuring the formation of single-strand DNA (ssDNA) gaps in U2OS WT and *PNKP*$^{-/-}$ C1 cells with 10 μM FEN1i and/or 10 μM PARP inhibitor (PARPi) treatment followed by S1 nuclease treatment. The ratio of CldU/IdU tract lengths in dual-labeled DNA fibers in the indicated cell lines and treatments were measured. Error bars represent SD. (**D**) Schematic and quantified results of experiments for measuring the formation of ssDNA gaps in U2OS WT and *PNKP*$^{-/-}$ C1 cells with 10 μM PARPi treatment followed by S1 nuclease treatment. The ratio of IdU/CldU tract lengths in dual-labeled DNA fibers in the indicated cell lines and treatments were measured.

*Figure 2 continued on next page*

*Figure 2 continued*

(**E**) Model of S1 nuclease-mediated digestion of DNA fiber in FEN1i- and/or PARPi-treated cells. In all panels, scale bar indicates 10 µm. Statistical significance was indicated as not significant (ns), *: $0.01< p \leqq 0.05$, **: $0.005 < p \leqq 0.01$ and ****: $0.0005 < p \leqq 0.001$.

The online version of this article includes the following figure supplement(s) for figure 2:

**Figure supplement 1.** FEN1 inhibitor treatment leads to faster fork speed.

PNKP deletion mutants (D1: FHA domain, D2: linker region, D3: phosphatase domain, and D4: kinase domain) (*Figure 3A–C*, *Figure 3—figure supplement 1A*; *Tsukada et al., 2021*). D2 mutant-expressing cells showed slower proliferation than cells expressing WT PNKP and other mutants, although D3 and D4 exhibited mildly slower cell proliferation (WT vs. D3: p = 0.1737; WT vs. D4: p = 0.4523). Furthermore, D3, D4 as well as D2 mutant-expressing cells showed increased tract lengths

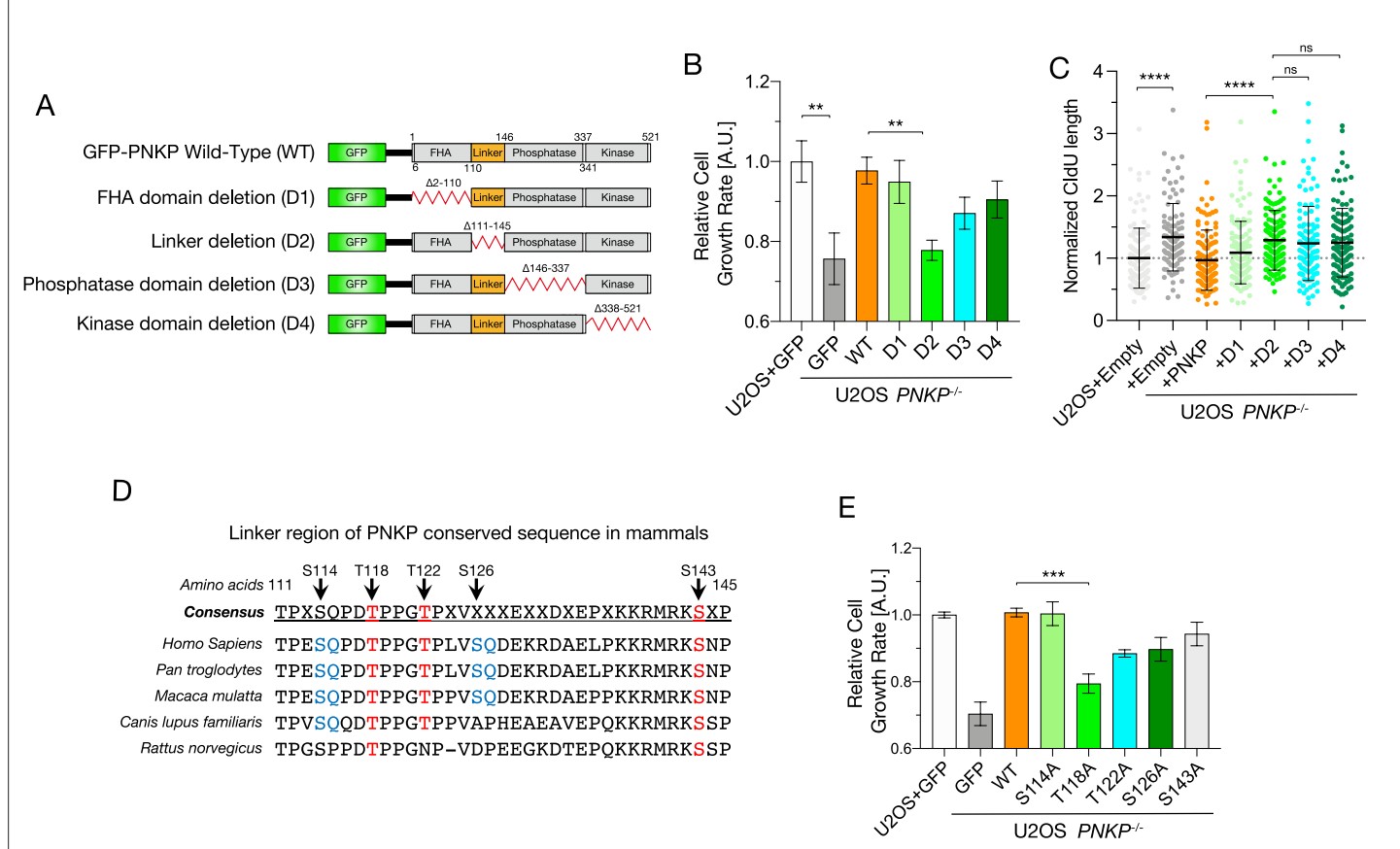

**Figure 3.** Phosphorylation of polynucleotide kinase phosphatase (PNKP), especially on T118, is required for cell proliferation and DNA replication. (**A**) Schematic diagrams of the structure of GFP-tagged human PNKP WT and deletion mutants (D1–D4). (**B**) Cell growth rate in U2OS WT and PNKP deletion mutant-expressing *PNKP*−/− C1 cells. Cell growth rate was normalized by GFP-expressing U2OS cells. Error bars represent SEM. (**C**) Quantified results of DNA fiber length in U2OS WT and *PNKP*−/− C1 cells expressing indicated PNKP deletion mutants. Nascent synthesized DNA was labeled by CldU. Error bars represent SD. (**D**) Alignment of amino acid sequences in linker region among mammalian species. SQ/TQ motif (blue) is PI3 kinase substrate motif. TP (red) is cyclin-dependent kinase (CDK) substrate motif. (**E**) Cell growth rate in U2OS WT and *PNKP*−/− C1 cells expressing indicated point mutants. Cell growth rate was normalized by GFP-expressing U2OS cells. Error bars represent SEM. Statistical significance was indicated as not significant (ns), **: $0.005< p \leqq 0.01$, ***: $0.001 < p \leqq 0.005$ and ****: $0.0005 < p \leqq 0.001$.

The online version of this article includes the following source data and figure supplement(s) for figure 3:

**Figure supplement 1.** Protein expression of polynucleotide kinase phosphatase (PNKP) mutants in U2OS cells.

**Figure supplement 1—source data 1.** Original membranes corresponding to *Figure 3—figure supplement 1A*.

**Figure supplement 1—source data 2.** Original membranes corresponding to *Figure 3—figure supplement 1A*.

**Figure supplement 1—source data 3.** Original membranes corresponding to *Figure 3—figure supplement 1B*.

**Figure supplement 1—source data 4.** Original membranes corresponding to *Figure 3—figure supplement 1B*.

compared to WT and D1 mutant-expressing cells, indicating that in addition to the enzymatic activity of PNKP, the linker region also plays a crucial role in proper fork progression.

Since these results indicate the linker region of PNKP is involved in proper fork progression, we attempted to identify essential amino acids for DNA replication within this region. Using Phospho-SitePlus, we identified five potential phosphorylation sites: serine 114, threonine118, threonine 122, serine 126, and serine 143 (*Figure 3D*; *Hornbeck et al., 2015*). Amino acids S114, T118, and S143 are highly conserved among mammalian species. S114 and S126 form a typical SQ/TQ motif, with S114 being phosphorylated by ATM and S126 by ATM and DNA-PKcs (*Segal-Raz et al., 2011*; *Zolner et al., 2011*). T118, T122, and S143 are novel phosphorylation sites, with T118 predicted to be a CDK phosphorylation substrate motif (S/TP). To elucidate the importance of these amino acids in DNA replication, we constructed phosphorylation-mutant vectors containing five predicted phosphorylated amino acids substituted with alanine and measured the growth rates of these transfectants (*Figure 3*, *Figure 3—figure supplement 1B, C*). T118A mutant-expressing cells exhibited a marked delay in cell growth, which was not observed in S114A, although T122A (p = 0.0258), S126A (p = 0.0523), and S143A (p = 0.4402) showed slight delays. These results suggest that the linker region of PNKP, especially the phosphorylation of PNKP on T118, is required for proper cell proliferation.

## CDKs phosphorylate T118 of PNKP and pT118-PNKP interacts with nascent DNA on replication forks

To assess the importance of T118 phosphorylation in DNA replication, we generated antibodies that recognized the phosphorylated T118 (pT118) peptides. The sufficient titers and specificities of the pT118 antibody were confirmed using ELISA (*Figure 4—figure supplement 1A, B*). The pT118 PNKP antibody was used for western blotting of lysates from U2OS WT-, green fluorescent protein (GFP)-PNKP WT-, or T118A-expressing cells. Although this antibody cross-reacted with proteins of approximately 55 kDa, which is close to the apparent molecular mass of endogenous PNKP, it clearly recognized GFP-tagged PNKP (GFP-PNKP) but not GFP-PNKP T118A (*Figure 4—figure supplement 1C*). Therefore, GFP-PNKP expression was used to examine T118 phosphorylation.

To determine whether T118 phosphorylation is DNA replication-specific, we synchronized HCT116 cells transiently expressing GFP-PNKP using a double thymidine block and released them at specific times (*Figure 4—figure supplement 1D*). After synchronization at the indicated cell cycle phases, cells were extracted and used for western blotting (*Figure 4A*). Since Cyclin A2 peaks during mid-to-late S/G2 phase and Cyclin E1 peaks at early S phase, we used these proteins as cell cycle markers (*Fung et al., 2007*; *Pagano et al., 1992*; *Honda et al., 2005*). pT118-PNKP was detected in asynchronized cells but increased particularly in the S phase, similar to Cyclin A2 expression levels. However, the reduction of pT118, possibly due to dephosphorylation of T118, was not as robust as the reduction in Cyclin A2 expression levels at the 12 hr time point. This effect was very weak during mitosis, suggesting that T118 phosphorylation plays a specific role in the S phase.

Since amino acids around T118 contain a CDK substrate motif, we measured the direct phosphorylation activity using purified CDKs, cyclin, and PNKP, and detected them using a pT118 PNKP antibody (*Figure 4B*). CDK1/Cyclin A2 and CDK2/Cyclin A2 markedly phosphorylated PNKP, whereas CDK4/Cyclin D1 and CDK2/Cyclin E1 phosphorylated PNKP to a lesser extent, suggesting that CDK1/CyclinA2 and CDK2/CyclinA2 complexes are potential kinases of PNKP T118. We subsequently investigated the phosphorylation levels of PNKP at T118 under co-overexpression of Cyclin A2 and CDK1 or CDK2 (*Figure 4C*). Overexpression of both CDK2/Cyclin A2 and CDK1/Cyclin A2 showed an increased phosphorylation level at T118, supporting that these complexes are potential kinases of PNKP T118.

In line with this observation, we assessed protein interactions between PNKP and CDK1 or CDK2 using GFP-PNKP-pulldown assay (*Figure 4—figure supplement 1E*). PNKP T118A mutant showed a reduced amount of protein interaction with both CDK1 and CDK2, with CDK2 appearing as a stronger binding partner of PNKP. These results indicate that CDK1/2, especially CDK2, are the kinases of T118 on PNKP. To elucidate the role of PNKP T118 phosphorylation in DNA replication, we isolated proteins from nascent DNA using iPOND (isolation of proteins on nascent DNA) technique to confirm interactions between PNKP and nascent DNA on replication forks (*Figure 4—figure supplement 1F*). After 5-ethynyl-2'-deoxyuridine (EdU) incorporation for 20 min in GFP-PNKP-expressing HEK293 cells, proteins bound to nascent DNA were extracted and detected by western blotting (*Figure 4D*). We found that replication fork-associated proteins, including FEN1, RPA2, and PCNA, as well as WT-PNKP

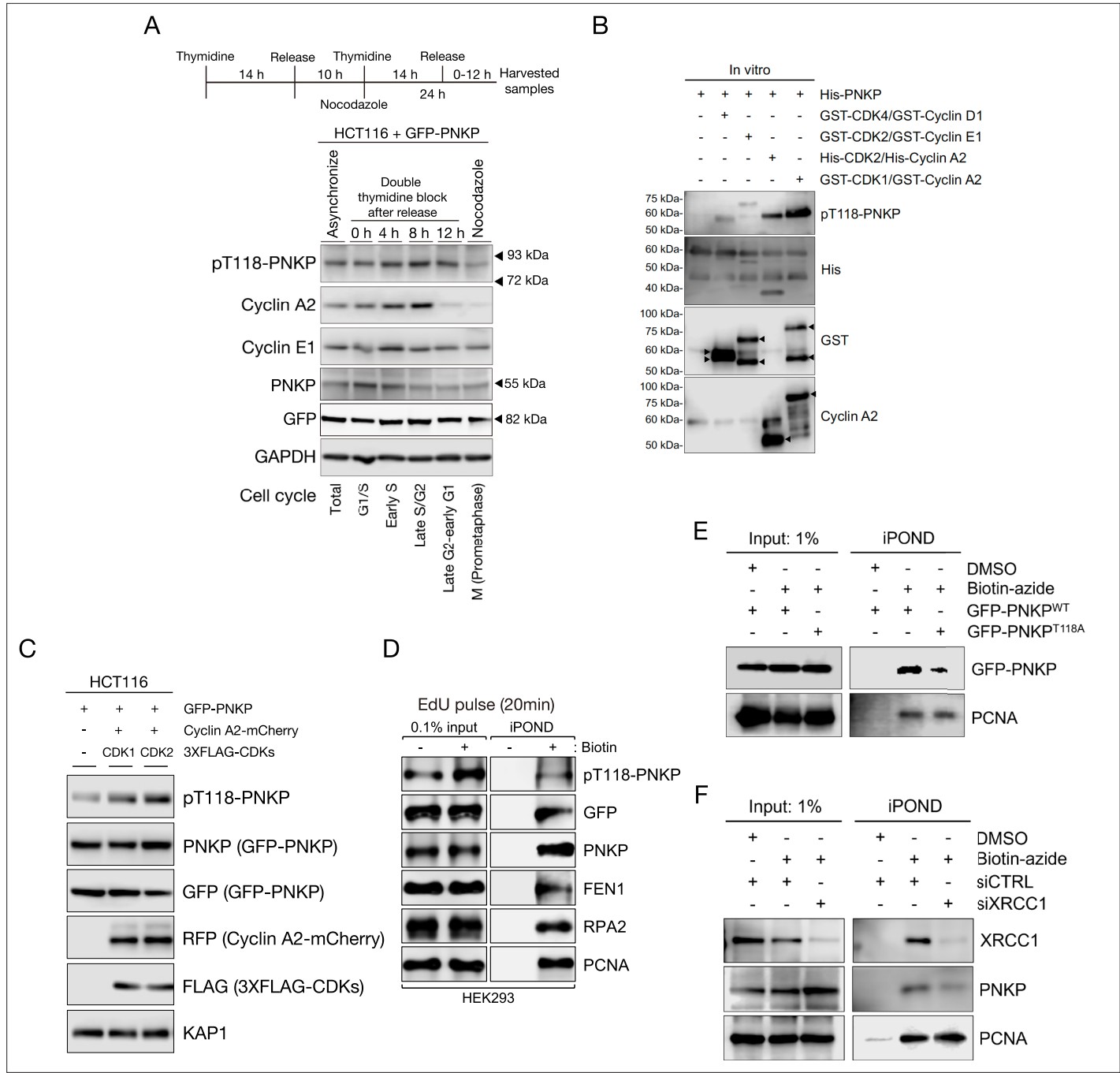

**Figure 4.** Cyclin-dependent kinases (CDKs) phosphorylate T118 on polynucleotide kinase phosphatase (PNKP) for the recruitment of PNKP to nascent DNA on replication forks. (**A**) Scheme and protein expression levels of GFP-PNKP-expressing HCT116 cells after release from double thymidine block. After released, cells were collected at indicated time points, and used for western blotting. pT118 PNKP antibody was generated for this study. Cyclin A2 and E1 antibodies were used for cell cycle markers. GAPDH antibody was used as a loading control. (**B**) In vitro analysis of PNKP phosphorylation on T118 by CDK/Cyclin complex. Purified His-PNKP and each CDK/Cyclin complex were incubated with reaction mixture and detected with western blotting by pT118-PNKP, His, GST, and Cyclin A2 antibodies. (**C**) HCT116 cells were lysed at 5 and 3 days after transfection with GFP-PNKP and co-transfection with mCherry2-Cyclin A2, and 3XFLAG-CDK1 or CDK2, respectively, and applied for western blotting. pT118 PNKP level was analyzed by pT118 PNKP-specific antibody. PNKP expression was analyzed with PNKP or GFP antibodies. Cyclin A2 expression was analyzed with RFP antibody. KAP1 antibody was used as loading control. (**D**) Analysis of isolated proteins from nascent DNA using isolation of proteins on nascent DNA (iPOND) technique. Proteins bound to EdU-labeled DNA in GFP-PNKP-expressing HEK293 cells were isolated using click reaction with biotin-azide followed by streptavidin-pulldowns and detected by western blotting. 0.1% of lysate used in Streptavidin-pulldowns represented as 0.1% input. (**E**) Western blotting

*Figure 4 continued on next page*

*Figure 4 continued*

to show interactions of GFP-PNKP WT or T118A with nascent DNA using the iPOND technique. PCNA was used as a loading control. (**F**) Western blotting to show interaction of PNKP with nascent DNA in XRCC1-depleted cells using the iPOND technique. PCNA was used as a loading control.

The online version of this article includes the following source data and figure supplement(s) for figure 4:

**Source data 1.** Original membranes corresponding to *Figure 4A*.

**Source data 2.** Original membranes corresponding to *Figure 4A*.

**Source data 3.** Original membranes corresponding to *Figure 4B*.

**Source data 4.** Original membranes corresponding to *Figure 4B*.

**Source data 5.** Original membranes corresponding to *Figure 4C*.

**Source data 6.** Original membranes corresponding to *Figure 4C*.

**Source data 7.** Original membranes corresponding to *Figure 4D*.

**Source data 8.** Original membranes corresponding to *Figure 4D*.

**Source data 9.** Original membranes corresponding to *Figure 4E*.

**Source data 10.** Original membranes corresponding to *Figure 4E*.

**Source data 11.** Original membranes corresponding to *Figure 4F*.

**Source data 12.** Original membranes corresponding to *Figure 4F*.

**Figure supplement 1.** Cyclin-dependent kinase (CDK)-mediated phosphorylation of polynucleotide kinase phosphatase (PNKP) on T118.

**Figure supplement 1—source data 1.** Original membranes corresponding to *Figure 4—figure supplement 1C*.

**Figure supplement 1—source data 2.** Original membranes corresponding to *Figure 4—figure supplement 1C*.

**Figure supplement 1—source data 3.** Original membranes corresponding to *Figure 4—figure supplement 1E*.

**Figure supplement 1—source data 4.** Original membranes corresponding to *Figure 4—figure supplement 1E*.

**Figure supplement 1—source data 5.** Original membranes corresponding to *Figure 4—figure supplement 1F*.

**Figure supplement 1—source data 6.** Original membranes corresponding to *Figure 4—figure supplement 1F*.

and T118-phosphorylated PNKP interacted with nascent DNA. Additionally, the recruitment of PNKP to nascent DNA was reduced by the T118A mutation, which strongly suggests T118 phosphorylation is crucial for its recruitment to replication forks (*Figure 4E*).

Since XRCC1 is a scaffold protein of ssDNA gap-filling pathway, we investigated whether the recruitment of PNKP to nascent DNA is dependent on XRCC1 using the iPOND technique (*Figure 4F*). Depletion of XRCC1 by siRNA treatment reduced the amount of PNKP recruited to nascent DNA, indicating the involvement of PNKP in ssDNA gap-filling pathway. However, some PNKP still bound to nascent DNA, suggesting an alternative recruitment pathway of PNKP to DNA replication forks, possibly for mediating canonical OF maturation. Taken together, these results suggest that the CDK1/Cyclin A2 or CDK2/Cyclin A2 complex potentially regulates the phosphorylation level of PNKP T118 in the S phase, facilitating PNKP recruitment to DNA replication forks.

## Phosphorylation of PNKP at T118 is required for preventing the formation of unligated OFs

In order to examine the effect of T118 phosphorylation loss in DNA replication, we first investigated the FEN1-related phenotype of increased tract length in T118A- and T118D (a phospho-mimetic mutant)-expressing cells using a DNA fiber assay (*Figure 5A*, *Figure 5—figure supplement 1A*). *PNKP*$^{-/-}$ and T118A-expressing cells showed increased tract length, and FEN1i treatment did not further accelerate DNA fiber length, indicating that the T118A mutation alone provokes the longer DNA tract length phenotype due to the formation of unligated OFs, similar to FEN1 inhibition. Conversely, T118D-expressing cells exhibited normal DNA fiber tract length, similar to PNKP WT-expressing cells, and FEN1i treatment increased DNA fiber tract length in T118D-expressing cells. This suggests that the T118 phosphorylation plays an essential role in proper OFs maturation akin to FEN1.

To specifically investigate the formation of ssDNA gaps in the S phase, we transiently expressed WT PNKP or T118A in *PNKP*$^{-/-}$ cells and incubated them with EdU-containing medium to separately assess ADP-ribose intensity inside (EdU-positive) and outside (EdU-negative) the S phase (*Figure 5B, C*). T118A-expressing cells in the S phase showed higher ADP-ribose intensity (1.61-fold increase vs.

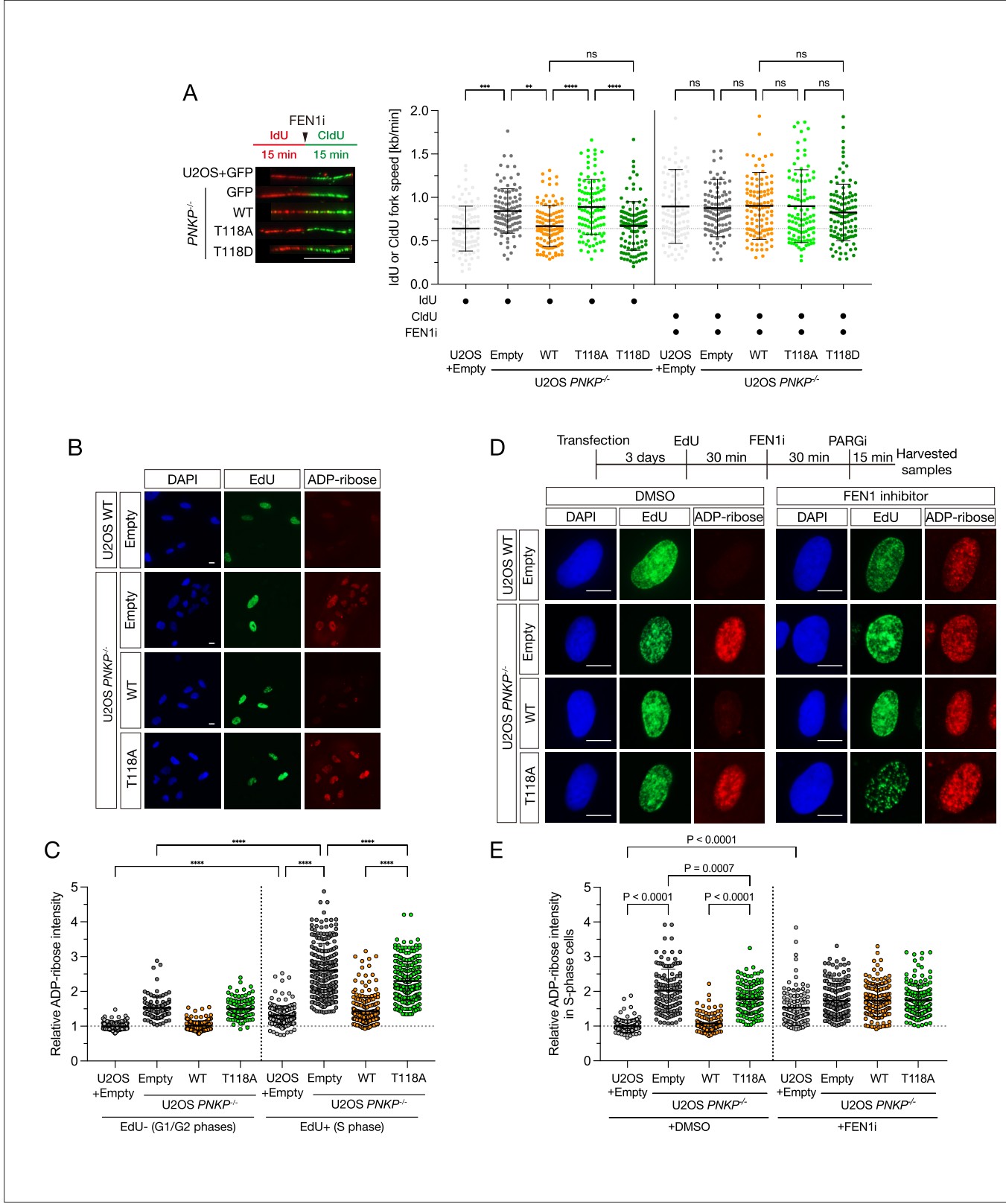

**Figure 5.** Phosphorylation of polynucleotide kinase phosphatase (PNKP) at T118 is required for preventing the formation of unligated Okazaki fragments. (**A**) Measurement of DNA synthesis speed in U2OS WT and PNKP$^{-/-}$ C1 cells expressing GFP, PNKP WT, T118A, or T118D mutants under 10 µM FEN1 inhibitor treatment analyzed by DNA fiber assay. At least 100 DNA fibers were measured. Error bars represent SD. (**B, C**) Representative images (**B**) and quantified results (**C**) of measurement of ADP-ribose intensity in U2OS WT and PNKP WT, PNKP T118-expressing *PNKP$^{-/-}$* C1 cells.

*Figure 5 continued on next page*

*Figure 5 continued*

Synthesized DNA was labeled by EdU and EdU-positive cells were defined as S phase and the other cells were defined as G1 +G2 phase. Error bars represent SEM. (D, E) Scheme, representative images (D) and quantified results (E) of the experiments for measurement of ADP-ribose intensity in U2OS WT and *PNKP*⁻/⁻ C1 cells expressing GFP, PNKP WT, and PNKP T118A mutant under DMSO (negative control) and FEN1i treatment only in EdU-positive (S phase) cells. Error bars represent SEM. In all panels, scale bar indicates 10 μm.

The online version of this article includes the following source data and figure supplement(s) for figure 5:

**Figure supplement 1.** Phosphorylation of polynucleotide kinase phosphatase (PNKP) on T118 is required for maintaining genomic stability.

**Figure supplement 1—source data 1.** Original membranes corresponding to *Figure 5—figure supplement 1A*.

**Figure supplement 1—source data 2.** Original membranes corresponding to *Figure 5—figure supplement 1A*.

WT-expressing cells), similar to *PNKP*⁻/⁻ cells (2-fold increase vs. U2OS WT cells). WT PNKP expression rescued this effect, suggesting that PNKP function and T118 phosphorylation are required for ssDNA gap-less DNA replication. Of note, *PNKP*⁻/⁻ cells and PNKP T118A cells also showed higher ADP-ribose intensity outside the S phase (1.47-fold increase vs. WT-expressing cells), indicating that PNKP and T118 may play a role in preventing SSBs formation outside the S phase. Since FEN1 has been reported to function in R-loop processing, PNKP could also be involved in this process (*Cristini et al., 2019*; *Laverde et al., 2022*). Future studies of a role of PNKP in different cell cycle will be able to address this question.

To investigate whether ssDNA gaps in T118A cells originate from unligated OFs, we used FEN1i to examine an epistatic effect between T118A and deficient canonical OFs maturation by FEN1 inhibition (*Figure 5D, E*). We assessed the ADP-ribose intensity of WT PNKP or T118A-expressing *PNKP*⁻/⁻ cells treated with either DMSO or FEN1i only during the S phase. FEN1i-treated U2OS WT cells showed increased ADP-ribose intensity, indicating that FEN1i treatment leads to ssDNA gap formation. The increased ADP-ribose intensity phenotype in T118A-expressing cells was not observed in FEN1i-treated T118A-expressing cells. Additionally, T118D-expressing cells behaved similarly to PNKP WT-expressing cells (*Figure 5—figure supplement 1B*). These data suggest that T118 phosphorylation of PNKP plays a critical role in suppressing ssDNA gap formation derived from unligated OFs in an epistatic pathway with FEN1. In conclusion, phosphorylation of PNKP at T118 is required to prevent unligated OF-mediated post-replicative ssDNA gap formation during DNA replication.

Incidentally, as similar to *PNKP*⁻/⁻ cells, PNKP T118A-expressing cells exhibited reduced repair abilities against exogenous SSB and DSB induced by IR and $H_2O_2$ treatment (*Figure 5—figure supplement 1C–F*). Furthermore, T118A-expressing cells showed an increased frequency of micronuclei and chromosome bridges compared to WT PNKP-complemented cells even without any treatment as well as with IR or HU treatment (*Figure 5—figure supplement 1G, H*). Altogether, our results suggest that endogenously increased ssDNA gap formation due to impaired OFs maturation in *PNKP*⁻/⁻ and T118A-expressing cells may cause genomic instability.

## Enzymatic activities of PNKP are important for the end-processing of OFs

The main function of PNKP is to catalyze 5'-phosphorylation and 3'-dephosphorylation of the DNA ends. To clarify the role of these enzymatic abilities in OFs maturation, we constructed phosphatase-dead (D171A) and kinase-dead (K378A) PNKP mutants (*Jilani et al., 1999*; *Reynolds et al., 2012*; *Kalasova et al., 2019*). Initially, we attempted to establish stable clones expressing both mutants; however, we could not obtain stable phosphatase-dead (D171A) clones. This is consistent with previous observations, where PNKP-mutated MCSZ patient cells, exhibiting mutations in the phosphatase domain, showed unstable expression of PNKP (*Shen et al., 2010*), and recombinant phosphatase-dead PNKP was unstable (*Kalasova et al., 2020*). Therefore, these mutants were transiently expressed in *PNKP*⁻/⁻ cells, and protein expression was assessed through western blotting (*Figure 6—figure supplement 1A*). We first assessed the phosphatase and kinase enzymatic activities of the PNKP T118A mutant and performed biochemical assays using cell extracts expressing PNKP mutants (*Figure 6—figure supplement 1B, C*). Fluorescence-labeled SSB gap oligo DNA was mixed with cell lysates extracted from U2OS WT or *PNKP*⁻/⁻ cells expressing various PNKP mutants. The D171A and K378A mutants were used as phosphatase- and kinase-dead controls, respectively. Although the phosphatase-dead mutant showed slightly lower kinase activity and the kinase-dead mutant showed lower phosphatase

activity than WT PNKP, it is possible that each mutant is structurally unstable, affecting enzyme activity. Intriguingly, T118A PNKP was still capable of dephosphorylating and phosphorylating gapped DNA ends in vitro, albeit to a lesser extent than WT PNKP. These results suggest that phosphorylation of PNKP at T118 is required for its recruitment to the gapped DNA structure but not directly for its enzymatic activity. To confirm this hypothesis, we assessed a binding ability to an ssDNA gap structure using cell lysates from WT PNKP, T118A, and T118D mutants (*Figure 6—figure supplement 1D*). The T118A mutant showed impaired gapped DNA-binding ability, whereas the WT PNKP and the T118D mutant exhibited relatively higher binding ability than the PNKP T118A mutant.

Next, we analyzed the increased tract length phenotype in cells expressing these enzymatic mutants using a DNA fiber assay (*Figure 6A*). Interestingly, D171A-expressing cells showed a spontaneously increased tract length, and FEN1i treatment did not further increase the tract length. K378A-expressing cells showed longer, yet statistically non-significant, tract length than WT PNKP-expressing cells (p = 0.6293), suggesting that PNKP enzymatic activities, especially those of phosphatase, are required for accurate fork progression. Subsequently, we elucidated whether the end-processing activities of PNKP are important for OF-mediated ssDNA gap formation. These PNKP mutant-expressing *PNKP⁻/⁻* cells were treated with FEN1i, followed by measurement of ADP-ribose intensity in the S phase (*Figure 6B, C*). D171A-expressing cells showed a high ADP-ribose intensity (p < 0.0001), while K378A-expressing cells exhibited relatively high ADP-ribose intensity without FEN1i treatment (p = 0.001). Moreover, FEN1i-treated cells showed high levels of ADP-ribose intensity in all conditions. Taken together, these results suggest that PNKP phosphatase and kinase activities, especially those of phosphatase, play an important role in the end-processing of OFs, resulting in the suppression of OF-mediated post-replicative ssDNA gap formation and ensuring accurate DNA replication (*Figure 6D*).

## Discussion

In this study, we identified the phosphorylation of PNKP at T118, mediated by CDKs, potentially by the CDK1/Cyclin A2 or CDK2/Cyclin A2 complex. This phosphorylation is crucial for the recruitment of PNKP to gapped DNA structures, including nicks between OFs and OF-mediated post-replicative ssDNA gaps (*Figure 6D*). Defects in PNKP phosphorylation at T118 lead to the accumulation of ssDNA gaps during DNA replication due to a deficiency in canonical OFs ligation and the subsequent gap-filling pathway.

PNKP consists of four regions: the FHA, linker, phosphatase, and kinase domains. Compared to other domains, the role of the linker region remains poorly understood, although it includes several residues that may be post-translationally modified. DNA damage signaling and DNA replication progression are often regulated by protein modifications such as phosphorylation and ubiquitination (*Kolas et al., 2007*; *Blackford and Jackson, 2017*; *Cortez et al., 1999*). Lysine is the main target of E3 ubiquitin ligase (*Laney and Hochstrasser, 1999*), and there is a clustered lysine region (137–142) in the PNKP linker region. However, this region also includes a nuclear localization signal, and alanine substitution prevents its transport to the nucleus (*Tsukada et al., 2020*). Therefore, we focused on regions other than amino acids 137–142. In the present study, we identified five predicted phosphorylation sites (S114, T118, T122, S126, and S143) in the linker region. The T118A mutant exhibited significantly reduced cell proliferation, while the T122A, S126A, and S143A mutants showed slightly reduced cell proliferation (*Figure 3E*), suggesting that these residues may also be involved in proper cell proliferation. In contrast, the S114A mutant-expressing cells showed effective SSB repair and normal cell growth (*Figure 3E*, *Figure 5—figure supplement 1*), indicating that the phosphorylation of PNKP at S114 is likely important for DSB repair. The T118A mutant demonstrated reduced repair abilities of both exogenous SSBs and DSBs (*Figure 5—figure supplement 1C–F*). Upon $H_2O_2$ treatment, T118A-expressing cells show a slight increase of ADP-ribosylation compared to WT-complemented cells, which might suggest a possible partial preferential role of this modification in the S phase. Future studies will be able to elucidate the function of T118 phosphorylation in response to exogenous DNA damage.

Regulating the speed of DNA synthesis is important for accurate DNA replication and fork integrity (*Kunkel, 2004*; *Genois et al., 2021*). *PNKP⁻/⁻* and T118A-expressing cells showed high-speed DNA synthesis, resulting in slower cell proliferation and genome instability (*Figures 1–3, 5, and 6*). These observations are consistent with those in PCNA KR mutant cells and PARPi-treated cells (*Thakar et al., 2020*; *Maya-Mendoza et al., 2018*). We also found that enzymatic activity, especially the phosphatase

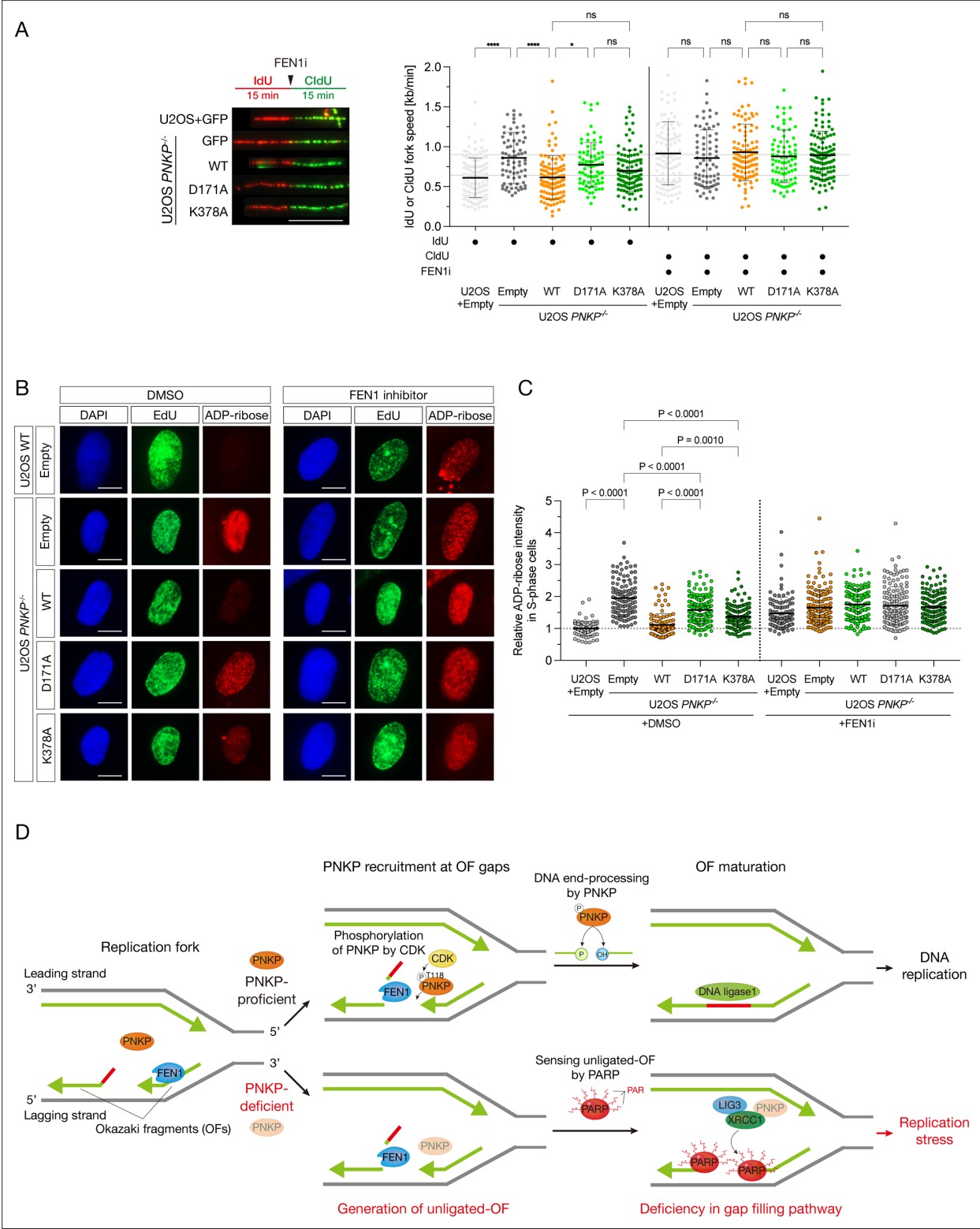

**Figure 6.** Enzymatic activities of PNKP is important for the end-processing of Okazaki fragments. (**A**) Measurement of the speed of DNA synthesis in U2OS WT and *PNKP*⁻/⁻ C1 cells expressing GFP, PNKP WT, D171A, and K378A mutants under 10 µM FEN1 inhibitor treatment analyzed by DNA fiber assay. At least 100 DNA fibers were measured. Error bars represent SD. Representative images (**B**) and quantified results (**C**) of the experiments for measurement of ADP-ribose intensity in U2OS WT and *PNKP*⁻/⁻ C1 cells expressing GFP, PNKP WT, PNKP D171A, and PNKP K378A mutants under DMSO (negative control) and FEN1i treatment only in EdU-positive (S phase) cells. Error bars represent SEM. (**D**) Model of the involvement of PNKP in

*Figure 6 continued on next page*

*Figure 6 continued*

DNA replication, especially in end-processing of canonical OFs ligation and gap-filling pathway. T118 of PNKP is phosphorylated by cyclin-dependent kinases (CDKs) for the recruitment to OFs during S phase. OFs ends are processed by PNKP and become the ligatable ends prior to the canonical OFs maturation. Unligated OFs are sensed by PARP1 for proceeding an alternative OFs maturation pathway, PARP-dependent gap filling, which requires proteins such as XRCC1, LIG3, and PNKP, to prevent the emergence of post-replicative single-strand DNA gaps. Impaired OFs maturation pathways lead to the accumulation of single-strand DNA gaps during DNA replication. In all panels, scale bar indicates 10 μm. Statistical significance was indicated as not significant (ns), *: $0.01 < p \leq 0.05$ and ****: $0.0005 < p \leq 0.001$.

The online version of this article includes the following source data and figure supplement(s) for figure 6:

**Figure supplement 1.** Enzymatic activity of PNKP T118A mutant.

**Figure supplement 1—source data 1.** Original membranes corresponding to *Figure 6—figure supplement 1A*.

**Figure supplement 1—source data 2.** Original membranes corresponding to *Figure 6—figure supplement 1A*.

**Figure supplement 1—source data 3.** Original gels corresponding to *Figure 6—figure supplement 1B*.

**Figure supplement 1—source data 4.** Original gels corresponding to *Figure 6—figure supplement 1B*.

**Figure supplement 1—source data 5.** Original gels corresponding to *Figure 6—figure supplement 1C*.

**Figure supplement 1—source data 6.** Original gels corresponding to *Figure 6—figure supplement 1C*.

activity of PNKP, is required for the end-processing of OFs during DNA replication (*Figure 6*). $PNKP^{-/-}$ and mutant-expressing cells exhibited a significant increase in ADP-ribose intensity in the S phase, even without FEN1i treatment. However, FEN1i treatment in these cells led to an attenuated increase in ADP-ribose intensity compared to WT PNKP-expressing cells, suggesting that PNKP acts in an epistatic pathway with FEN1 for the canonical OFs ligation pathway (*Figure 6B, C*). FEN1 catalyzes the removal of RNA/DNA fragments from OFs on the lagging strand. Our data suggested that both the end-processing activity of PNKP and the exonuclease activity of FEN1 are required for canonical OFs ligation (*Figures 2C, D, 5E, and 6C*).

Furthermore, the PARP-mediated ssDNA gap-filling pathway is also involved in OFs maturation as an alternative/backup pathway (*Vaitsiankova et al., 2022*). Our data suggest that the loss of these two pathways results in extensively increased ssDNA gap formation during DNA replication, and the loss of PNKP can recapitulate this combinative phenotype (*Figure 2C*). This observation is consistent with the decreased ADP-ribose intensity observed in HU- and EME-treated $PNKP^{-/-}$ cells (*Figure 1E*). Since EME inhibits single-strand gap formation (*Burhans et al., 1991*; *Lukac et al., 2022*) and HU treatment increases the amount of the OF-like DNA fragments (*Laipis and Levine, 1973*; *Magnusson, 1973b*; *Magnusson, 1973a*), these reagents inhibit mature OF structures as PNKP-appropriate substrates.

Moreover, we found that the phosphorylation of PNKP at T118 is regulated in a cell cycle-dependent and CDK-associated manner (*Figure 4A–C*, *Figure 4—figure supplement 1E*). Since the activities of CDK1/Cyclin A2 and CDK2/Cyclin A2 complexes peak in the late S/G2 and S phases, respectively, we speculated that PNKP T118 is phosphorylated from the S phase to the G2 phase in CDK1/Cyclin A2- and CDK2/Cyclin A2-dependent manner (*Figure 4B, C*). We concluded that phosphorylation of PNKP at T118 allows it to be recruited to the ends of OFs, including ssDNA nicks and/or gaps between OFs, and processes the ends for subsequent ligation. Furthermore, PNKP is required for the PARP1-dependent ssDNA gap-filling pathway when unligated OFs are transformed into post-replicative ssDNA gap structures (*Figure 2C*).

In summary, maintaining genome stability is central to life, and the function of PNKP in DNA repair and OFs maturation is important for biological development. Although PNKP mutations cause several inherited diseases (MCSZ, AOA4, and CMT2B2) with neural developmental failure and neurodegeneration, almost all mutations are found in the FHA, phosphatase, or kinase domains and not in the linker region. This observation may indicate that mutations in the phosphorylated residue (T118) in the linker region are potentially embryonic lethal due to the importance of T118 in DNA replication, which is revealed in the present study. Since the inhibiting PNKP activity is an important target for anticancer drugs, in addition to the inhibition of phosphatase and kinase enzymatic activity (*Freschauf et al., 2009*), inhibiting phosphorylation at T118 of PNKP might be a potent target for cancer therapy. Ultimately, this study reveals a novel role for PNKP in processing the ends of OFs and the PARP-dependent single-strand gap-filling pathway during DNA replication. This may contribute to the elucidation of the

biological basis of DNA replication and enhance our understanding of the mechanisms underlying the onset of inherited diseases.

## Materials and methods

### Cell culture

The human embryonic kidney cell line HEK293, the human colon cancer cell line HCT116, and the human osteosarcoma cell line U2OS were obtained from the American Type Culture Collection (ATCC), and U2OS $PNKP^{-/-}$ cell lines were established in this study. All cell lines were maintained in Dulbecco's modified Eagle's medium (Nacalai Tesque Inc) supplemented with 10% vol/vol fetal bovine serum (Hyclone, GE Healthcare) and penicillin/streptomycin (Nacalai Tesque Inc) at 37°C in humidified atmosphere containing 5% $CO_2$ conditions. All cell lines were tested for *Mycoplasma* contamination using the e-Myco Mycoplasma Detection PCR Kit (iNtRON Biotechnology, Inc, cat# 25235).

### Construction of plasmid DNA and mutagenesis

The pEGFP-C1 plasmid was purchased from Clontech. Full-length human PNKP cDNA was obtained by PCR from the cDNA pool of U2OS cells and inserted into the pEGFP-C1. Mutations were introduced using the PrimeSTAR mutagenesis basal kit (Takara Bio, cat# R046A) according to the manufacturer's instructions. All the DNA constructs were verified by DNA sequence analysis. All primers for mutagenesis were designed using the Agilent QuikChange primer design program and are listed in *Supplementary file 1, table 1*.

### cDNA and siRNA transfection

For cDNA transfection, PEI-MAX (Polysciences, Inc, cat# 24765) or Lipofectamine 3000 (Invitrogen, Thermo Fisher Scientific) were used according to the manufacturer's instructions. For siRNA transfection (treatment time is typically – 48–72 hr), Lipofectamine RNAiMAX (Invitrogen, Thermo Fisher Scientific) was used according to the manufacturer's instructions. All the siRNAs were used at a final concentration of 50 nM. The targeting sites and sequences of the siRNA oligonucleotides are listed in *Supplementary file 2, table 2*.

### Genome editing by CRISPR/Cas9 system and establishment of PNKP$^{-/-}$ cell lines

The pSpCas9n(BB)-2A-Puro (PX462) vector was purchased from Addgene. For the establishment of $PNKP^{-/-}$ cells using CRISPR/Cas9 system, the sgRNA target sequences were cloned into the pSpCas9n(BB)-2A-Puro (PX462) vector and verified by DNA sequencing. U2OS cells were transfected with the targeting vectors and incubated for 2 days before the addition of selective medium containing 1.5 µg/ml puromycin (InvivoGen, cat# ant-pr-1). After 5 days, clonal cells were isolated by limiting dilution in 96-well plates. PNKP expression in single clones was analyzed by western blotting. Genomic mutations in both PNKP alleles present in U2OS cells were verified by DNA sequencing of a PCR-amplified genomic fragment cloned into a pEGFP-C1 vector. The sgRNA target sequences are shown in *Figure 1—figure supplement 1*.

### Sodium dodecyl sulfate–polyacrylamide gel electrophoresis and western blotting

Cells were lysed in a radioimmunoprecipitation assay buffer (50 mM Tris-HCl, pH 8.0, 250 mM NaCl, 25 mM ethylenediaminetetraacetic acid [EDTA], 0.5% vol/vol Triton X-100, 0.5% wt/vol sodium dodecyl sulfate [SDS], and 0.5% wt/vol sodium deoxycholate) containing protease inhibitor cocktail (Nacalai Tesque Inc, cat# 25955-11) and phosphatase inhibitor cocktail (Nacalai Tesque Inc, cat# 07575-51), and the protein concentration was measured by a bicinchoninic acid (BCA) assay kit (Takara Bio) using bovine serum albumin (BSA) as the standard. In all experiments, 20 µg of protein was loaded onto SDS–polyacrylamide gel electrophoresis (PAGE) plates. The proteins were electrophoresed at 30 mA/gel plate for 1–1.5 hr and transferred onto a polyvinylidene fluoride (PVDF) membrane at 100 V for 1.5 hr. Next, the PVDF membrane was blocked with either 2% wt/vol BSA/TBS-T (tris-buffered saline and Tween 20) or 5% wt/vol skim milk/TBS-T for 1 hr at room temperature on a shaker. For primary antibody reactions, the following primary antibodies were used for 1–4 hr at room temperature: PNKP

(rabbit, 1:1000, Novus, cat# NBP1-87257), PNKP (rabbit, 1:1000, Abcam, cat# ab181107), pT118-PNKP (rabbit, 1:1000, generated in this paper), pS114-PNKP (rabbit, 1:1000, generated in this paper), XRCC1 (mouse, 1:1000, Invitrogen, Thermo Fisher Scientific, cat# MA5-13412), XRCC4 (rabbit, 1:1000, generated in our laboratory; *Kamdar and Matsumoto, 2010*), PCNA (rabbit, 1:500, Santa Cruz Biotechnology, cat# sc-7907), KAP1 (rabbit, 1:1000, abcam, cat# ab10483), pS824-KAP1 (rabbit, 1:1000, BETHYL, cat# A300-767A-2), Cyclin-A2 (mouse, 1:1000, Cell signaling, cat# BF683), Cyclin-E1 (rabbit, 1:1000, Sigma-Aldrich, cat# C4976), GFP (mouse, 1:3000, Nacali Tesque Inc, cat# GF200), ATM (mouse, 1:2000, Sigma-Aldrich, cat# A1106), DNA-PKcs (rabbit, 1:2000, abcam, cat# Y393), p53 (mouse, 1:5000, Santa Cruz Biotechnology, cat# sc-126), RPA2 (mouse, 1:3000, abcam, cat# ab2175), GAPDH (mouse, 1:10,000, EMD Millipore, cat# MAB374), FLAG-M2 (mouse, 1:1000, Sigma-Aldrich, cat# A8592), RFP (rabbit, 1:1000, MBL, PM005), and FEN1 (mouse, 1:500, Santa Cruz, sc-28355). The PVDF membrane was washed three times with TBS-T. For secondary antibody reactions, horseradish peroxidase (HRP)-conjugated rabbit or mouse antibodies (Dako, cat# P0399 or P0447, respectively) were incubated for 1 hr at room temperature. After washing five times with TBS-T, the membranes were developed using enhanced chemiluminescence (LI-COR Biosciences) and detected using a C-digit (LI-COR, Biosciences).

## Immunoprecipitation

For sample preparation for immunoprecipitation, HEK293 cells were grown on 100 mm dishes, washed twice in phosphate-buffered saline (PBS; Nacalai Tesque Inc), and lysed in lysis buffer (50 mM Tris-HCl, pH 7.5, 100 mM NaCl, 0.2% NP-40, 1 mM MgCl$_2$, and 10% glycerol) supplemented with cocktails of protease inhibitors and phosphatase inhibitors. After incubation for 30 min on the rotator at 4°C, lysates were cleared by centrifugation 20,000 × *g* for 20 min at 4°C. Next, lysates were incubated with 10 μl of GFP-Trap magnetic agarose beads (ChromoTek, GmbH) for 4 hr with mixing on a rotator at 4°C. The beads were then washed five times with lysis buffer, and proteins were eluted in 2× SDS sample buffer (125 mM Tris-HCl, pH 6.8, 4% wt/vol SDS, 20% vol/vol glycerol, 0.01% wt/vol bromophenol blue, and 5% vol/vol 2-mercaptoethanol).

## Immunofluorescence

Cells were grown on glass coverslips and fixed with 4% formaldehyde for 15 min at 4°C. Cells were subsequently permeabilized with PBS containing 0.2% Triton X-100 for 5 min at 4°C. Following 30 min of blocking in PBS supplemented with 2% wt/vol BSA, primary antibody reactions were performed in PBS-T supplemented with 1% wt/vol BSA for 2 hr at room temperature. Cells were washed three times with PBS, and secondary antibody reactions were performed in PBS-T supplemented with 1% BSA for 1 hr at room temperature in the dark. After washing five times with PBS, coverslips were mounted in mounting medium (Dako) containing the nuclear staining dye 4',6-diamidino-2-phenylindole dihydrochloride (DAPI) and allowed to dry for 2 hr at room temperature in the dark. For the primary antibody, an anti-γH2AX mouse antibody (Merck Millipore, JBW301) was used at a 1:1000 dilution. An Alexa Fluor 594-conjugated mouse secondary antibody (Invitrogen, Thermo Fisher Scientific, cat# A32742) was used at a 1:2000 dilution.

To quantify γH2AX foci formation, nuclei and foci-positive cells were counted using the ImageJ software. Foci-positive cells were defined as those containing >10 foci, and at least 100 cells were counted. Representative images are presented.

To measure ADP-ribosylation, cells were pretreated with 10 μM poly (ADP-ribose) glycohydrolase inhibitor (PARGi, TOCRIS Bio-Techne, PDD 00017273) for 30 min prior to IR exposure to increase total ADP-ribosylation levels by inhibiting hydrolysis of the ribose-ribose bonds present in poly (ADP-ribose). For the HU and EME assays, cells were incubated in 2 mM HU (Sigma-Aldrich) for 2 hr or in 1 μM EME (Bio vision, BVN-B2339-50-50) for 1 hr, with PARG inhibitor added during the final 20 min. For the H$_2$O$_2$ assay, cells were labeled with 10 μM EdU for 20 min and then treated with 1 mM H$_2$O$_2$ for 10 min. As the primary reaction to detect SSBs, a PAN ADP-ribose-binding reagent (rabbit, Merck, cat# 9QQ12P) was used at 1:1000 dilution in PBS-T supplemented with 1% wt/vol BSA. Alexa Fluor 488- or 594-conjugated rabbit secondary antibodies (Invitrogen, cat# A32731 or A32740, respectively) was used at a 1:2000 dilution. EdU labeling was performed between blocking and primary antibody reaction using Click-iT EdU Alexa Fluor 488 Imaging Kit (Life Technologies, cat# C10337) according to the manufacturer's instructions. The mean intensity of ADP-ribose in nuclei stained with

DAPI was measured using ImageJ software. At least 100 cells were counted, and the average ADP-ribose intensity was calculated using GraphPad Prism 8 (GraphPad Software Inc).

To measure genome instability, cells with micronuclei and chromosome bridges were counted using DAPI staining. At least 300 cells were counted using ImageJ software.

### BrdU incorporation assay

Cells were grown on glass coverslips and labeled with 10 μM BrdU (Sigma) for 48 hr. After treatment with 100 μM $H_2O_2$ for 10 min, cells were pre-extracted with CSK buffer (10 mM PIPES-NaOH, pH 6.8, 100 mM NaCl, 300 mM sucrose, 3 mM $MgCl_2$, and 1 mM EGTA) containing 0.5% (vol/vol) Triton X-100 for 10 min at 4°C and then fixed with 2% (wt/vol) formaldehyde containing 0.2% (vol/vol) Triton X-100 for 10 min at room temperature followed by wash twice with PBS-T. Following treatment with 0.1% (wt/vol) pepsin (Tokyo Chemical Industry) in 10 mM HCl for 10 min at 37°C, cells were washed twice with ExoIII buffer (50 mM Tris-HCl, pH 7.5, 10 mM $MgCl_2$, 1 mM DTT) and then incubated in ExoIII buffer containing 10 u/ml ExoIII (Promega) for 15 min at 37°C. After quench by adding 20 mM EDTA and block with PBS-T containing 10% (vol/vol) new born calf serum (nBCS; Hyclone, GE Healthcare), cells were reacted with anti-BrdU (mouse, 1/100, BD Biosciences, cat# 347580) in PBS-T containing 10% (vol/vol) nBCS overnight at 4°C. Following wash twice with PBS-T, secondary antibody reaction was performed in PBS-T containing 10% (vol/vol) nBCS for 1 hr at room temperature in the dark. After washing five times with PBS-T, DAPI stain was performed in PBS-T on a shaker for 30 min at room temperature. Afterwards, coverslips were mounted in mounting medium and then dried overnight at room temperature in the dark followed by microscopy. The mean intensity of native BrdU in nuclei stained with DAPI was measured using CellProfiler (*Stirling et al., 2021*) software. At least 400 cells were counted, and the average of native BrdU intensity was calculated using GraphPad Prism 8 (GraphPad Software Inc).

### γ-Ray irradiation

To examine the sensitivity and response to IR, cells were irradiated using (*Kamdar and Matsumoto, 2010*) Co γ-ray source in Tokyo Institute of Technology. The dose rate was measured using an ionizing chamber-type exposure dosimeter C-110 (Oyo Giken, Tokyo, Japan) and corrected for decay.

### Colony formation assay

The surviving fraction was determined using the colony formation assay. The cells were plated on 100 mm dishes. The number of plated cells was adjusted using higher doses of the indicated DNA-damaging agents to obtain an appropriate number of colonies. After incubation for 12–14 hr at 37°C under 5% $CO_2$ conditions, cells were exposed to grading doses of γ-ray (1, 3, and 5 Gy), $H_2O_2$ (100 and 200 μM for 2 hr), and HU (1, 2, and 4 mM for 24 hr). The cells were further incubated for 10–14 days to form colonies. After washing with PBS, cells were fixed with 99.5% ethanol and stained with staining solution (0.02% wt/vol crystal violet; 2.5% vol/vol methanol). After washing the plates twice with water and drying overnight, colonies containing more than 50 cells were counted manually. The plating efficiency was calculated as the number of colonies divided by the number of plated cells. The surviving fraction was calculated as the plating efficiency of the irradiated cells divided by the plating efficiency of the unirradiated cells. Experiments were independently repeated at least three times.

### Cell growth assay

Cell growth and growth rates were analyzed based on the number of cells at several time points (*Figure 1B*: days 0, 1, 2, 3, and 4; *Figure 3B, E*: days 0, 1, and 4). Cells were prepared at 70–80% confluency on 100 mm dishes or 60 mm dishes and spread onto 6-well plates ($2 \times 10^5$ cells/well). The cells were cultured at 37°C in humidified atmosphere containing 5% $CO_2$ conditions. After the indicated incubation period, cells were harvested by trypsinization, and cell numbers were counted using a Coulter counter (Beckman Coulter) in all experiments.

### Cell cycle distribution analysis by flowcytometry

The procedure for cell cycle distribution analysis has been described in our recent publication (*Tsuchiya et al., 2021*) and was appropriately modified for this study. In brief, the nascently synthesized DNA was labeled with EdU and Alexa Fluor 488 azide through Click reaction using a Click-iT EdU Imaging

kit (Life Technologies, cat# C10337), and cells were then stained with propidium iodide (PI) using a Cell Cycle Phase Determination Kit (Cayman Chemicals, cat# 10009349) according to the manufacturer's instruction. Cells grown in 6-well plates or 60 mm dish at 70–90% confluency were treated with 10 µM of EdU for 1 hr and harvested by trypsinization. Harvested cells were washed with 0.1% wt/vol BSA/PBS and fixed/permeabilized by BD cytofix/cytoperm buffer (BD Biosciences, cat# 554714) according to the manufacturer's instructions. Subsequently, the cells were washed with 1× BD perm/wash buffer and resuspended in a click-it reaction cocktail for 1 hr at room temperature in the dark. After Click reaction, cells were washed with 1× BD perm/wash buffer and resuspend in PBS containing 0.02% wt/vol sodium azide, 0.02% wt/vol RNaseA, and 0.01% wt/vol PI for 1 hr at room temperature in the dark. The cell suspension was supplemented with 500 ml of 0.1% wt/vol BSA/PBS for adjustment of the volume prior to the analysis and subjected to flow cytometry using Cell Lab Quanta SC (Beckman Coulter).

## DNA fiber analysis

The DNA fiber assay was performed according to a previously reported paper (*Schwab and Niedzwiedz, 2011*) and appropriately modified for this study. Cells grown on 60 mm dish at 70–90% confluency were initially labeled with 50 µM IdU for 15 min and subsequently labeled with 250 µM CldU for 15 min in a humidified $CO_2$ incubator. Labeled cells were harvested by trypsinization and resuspended in ice-cold PBS at $1 \times 10^6$ to $1 \times 10^7$ cells/ml.

For the S1 nuclease assay, cells were labeled with 250 µM of CldU for 60 min with 10 µM of FEN1 inhibitor (MedChemExpress, FEN1-IN-3) and/or treated with 10 µM of PARP inhibitor (AdooQ Bioscience, Olaparib, AZD2281) in a humidified $CO_2$ incubator, washed once with PBS, and permeabilized with CSK-100 buffer (100 mM NaCl, 10 mM HEPES pH 7.8, 3 mM $MgCl_2$, 300 mM sucrose, and 0.5% Triton X-100) for 10 min at RT. After permeabilization, cells were washed twice with ice-cold PBS and harvested by scraping. Harvested cells were divided into two 1.5 ml tubes for S1 nuclease treatment and subsequently centrifuged at $400 \times g$ for 3 min. Cell pellets were treated with S1 nuclease (Takara Bio, 2410A) for 30 min at 37°C and resuspended in ice-cold PBS at $1 \times 10^6$ to $1 \times 10^7$ cells/ml.

Two µl of the cell suspension was spotted on one end of the glass slides (Matsunami glass, cat# S8215) and air-dried for 5 min. 7 µl of DNA fiber lysis buffer (200 mM Tris-HCl, pH 7.5, 50 mM EDTA, and 0.5% wt/vol SDS) were added to the cell suspension, gently stirred with a pipette tip, and incubated for 2 min. The glass slides were tilted at 15° to allow the fibers to spread along the slide and air-dried once the fiber solution reached the end of the glass slide. Glass slides were immersed in fixative (75% vol/vol methanol and 25% vol/vol acetic acid) and incubated for 10 min. After washing with distilled water twice, glass slides were immersed in 2.5 M HCl for 80 min, followed by three times wash with PBS, and blocked with 5% wt/vol BSA/PBS for 30 min. For primary antibody reaction, anti-BrdU (mouse, 1/100, BD Biosciences, cat# 347580, reacts with IdU) and anti-BrdU (rat, 1/400, abcam, cat# ab6326, reacts with CldU) diluted in 5% wt/vol BSA/PBS were used and incubated in a humidified case for 2 hr. After washing with PBS three times, goat anti-rat Alexa Fluor 488 (Invitrogen, 1/1000, cat# A110060) and goat anti-mouse Alexa Fluor 594 (Invitrogen, 1/1000, cat# A11005) were put onto the glass slides for secondary antibody reaction and incubated for 1 hr in the dark. The glass slides were subsequently washed three times with PBS-T and mounted using mounting medium (Dako). To observe DNA fibers, an OLYMPUS IX71 (OLYMPUS) or Zeiss LSM880 (Carl Zeiss) fluorescence microscope was used, and at least 50 fibers were measured in each experiment. The tract length of the DNA fibers was measured using the ImageJ software and analyzed using GraphPad Prism 8 (GraphPad Software Inc).

## Protein–DNA-binding assay

An EpiQuik Colorimetric General Protein-DNA Binding Assay Kit (Epigentek Inc, cat# P-2004-96) was used to measure the DNA-binding ability of WT, T118A, and T118D PNKP, according to the manufacturer's instructions. Nuclear extracts were harvested from U2OS *PNKP*$^{-/-}$ cells transfected with WT GFP-PNKP, T118A, or T118D expression vectors in non-denaturing lysis buffer (150 mM KCl, 50 mM Tris-HCl, pH 8.3, 1 mM EDTA, and 1 mM DTT) supplemented with protease inhibitors. To measure the DNA-binding ability to the gapped DNA, a biotinylated oligonucleotide (BioF20: 5′-Biotin/TAGC ACCTACCGATTGTATG/Phos-3′) and a non-biotinylated oligonucleotide (F15: 5′/TACGTTTTTGTG TCG/3′) were annealed to a complementary strand oligonucleotide (R36: 5′-Phos/CGACACAAAAAC

GTATCATACAATCGGTAGGTGCTA/3′) in annealing buffer (10 mM Tris, pH 7.5, 50 mM NaCl, and 1 mM EDTA). Oligonucleotides in annealing buffer were incubated at 95°C for 5 min and cooled down slowly. 20 ng of biotinylated double-stranded oligonucleotides and 10 µg of nuclear extract was used for the DNA-binding reaction in streptavidin-coated tubes. Additionally, 1 µg/ml GFP antibody (Nacali Tesque Inc, cat# GF200) and 0.5 µg/ml HRP-conjugated mouse antibody (Dako, cat# P0447) were used to detect DNA-binding proteins. The absorbance was measured at 450 nm using an iMark Microplate Absorbance Reader (Bio-Rad Laboratories).

## PNKP phosphatase and kinase activity biochemical assay

The phosphatase and kinase activities of PNKP were determined in accordance with previous studies (*Kalasova et al., 2020*; *Dobson and Allinson, 2006*). Fluorescently labeled oligonucleotides (Integrated DNA Technologies) were used as PNKP substrates. For 3′-phosphatase reaction, 'S1' [5′-(TAMRA) TAGCATCGATCAGTCCTC-3′-P] and 'C2' [5′-P-GAGGTCTAGCATCGTTAGTCA-(6-FAM)-3′] were annealed to a complementary strand oligonucleotide 'B1' [5′-TGACTAACGATGCTAGACCTCTGA GGACTGATCGATGCTA-3′] in annealing buffer (10 mM Tris pH 7.5, 200 mM NaCl, and 1 mM EDTA). For 5′-kinase reaction, 'C1' [5′-(TAMRA)-TAGCATCGATCAGTCCTC-3′-OH] and 'S2' [5′-OH-GAGG TCTAGCATCGTTAGTCA-(6-FAM)-3′] were annealed to 'B1' oligonucleotides in annealing buffer. Each of these oligonucleotide mixtures was treated at 95°C for 5 min and then left at room temperature for 1 hr to anneal and form substrate oligonucleotides. The expression vectors for WT PNKP, T118A, D171A, and K378A were transfected into $PNKP^{-/-}$ cells. In addition to these transfected $PNKP^{-/-}$ cells, WT U2OS and $PNKP^{-/-}$ cells were suspended in lysis buffer (25 mM Tris, pH 7.5, 10 mM EDTA, 10 mM EGTA, 100 mM NaCl, and 1% Triton X-100) and incubated for 15 min at 4°C and then centrifuged at 16,000 × *g* for 20 min at 4°C. The supernatant was used as the cell-free protein extract. Protein extract of $1 \times 10^5$ cells was incubated with 100 nM substrate oligonucleotides and 20 µM single-stranded nuclease competitor oligonucleotide [5′-AAAGATCACAAGCATAAAGAGACAGG-3′] in reaction buffer (25 mM Tris, pH 7.5, 130 mM KCl, 10 mM MgCl2, 1 mM DTT, and 1 mM ATP) for 10 min at 37°C. The enzymatic reactions were terminated by adding 25 µl of quenching buffer (90% formamide, 50 mM EDTA, 0.006% Orange G) to 25 µl of reaction solution. Each reaction sample was diluted with quenching buffer (10×). Subsequently, 10 µl of each reaction sample was separated on a 20% denaturing polyacrylamide gel (7 M Urea and TBE buffer) for 16 hr (500 V, 10 mA) and analyzed on a Typhoon 9500 (GE Healthcare Life Science).

## Isolation of proteins on nascent DNA

iPOND experiments were performed according to a previous protocol paper (*Dungrawala and Cortez, 2015*) and appropriately modified for this study. HEK293 cells grown in 15 cm dishes were treated with 10 µM EdU for 10 min. After EdU labeling, cells were fixed in 10 ml of 1% formaldehyde/PBS on the dishes for 20 min at room temperature and then quenched by adding 1 ml of 1.25 M glycine. Cells were harvested by scraping 5 min after quenching and washed three times with PBS. Cells were subsequently permeabilized with PBS containing 0.25% Triton X-100 for 30 min at room temperature and washed twice with PBS. Each sample was divided into two 1.5 ml tubes for a click reaction (with biotin-azide) and a no-click control (without biotin-azide). Click-iT Reaction Buffer (Thermo Fisher, cat#: C10269) and biotin-azide (Cayman Chemical, cat#: 13040) were used for the Click reaction, according to the manufacturer's protocol. Cells were washed twice in PBS and subsequently lysed in iPOND lysis buffer (1% SDS and 50 mM Tris-HCl pH8.0) containing protease inhibitor cocktail. Samples were sonicated using a BRANSON 150 sonicator, centrifuged at 20,000 × *g* for 10 min at room temperature, and diluted in a 1:1 volume of PBS containing a protease inhibitor cocktail. 20 µl (per sample) of Streptavidin-Magnetic beads (Thermo Fisher, cat#: 88816) or 100 µl (per sample) of Dynabeads MyOne Streptavidin C1 (Thermo Fisher, cat#: 65001) were washed twice with iPOND lysis buffer and incubated with samples overnight at 4°C in the dark. Bead-sample mixtures were washed once in iPOND lysis buffer, once with low salt buffer (1% Triton X-100, 2 mM Tris-HCl pH 8.0, 2 mM EDTA, and 150 mM NaCl), once with high salt buffer (1% Triton X-100, 2 mM Tris-HCl pH 8.0, 2 mM EDTA, and 500 mM NaCl), and once with iPOND lysis buffer. Proteins were eluted in 2× SDS sample buffer by incubating for 25 min at 95°C. Samples were resolved on SDS–PAGE, and proteins were detected by immunoblotting.

## In vitro kinase assay

Purified human recombinant 6×His-tagged PNKP was generously provided by Dr. Michael Weinfeld. Purified human recombinant GST-tagged CDK4/Cyclin D1 (cat# PV4400), CDK2/Cyclin E1 (cat# PV6295), CDK1/Cyclin A2 (cat# PV6280), and 6×His-tagged CDK2/Cyclin A2 (cat# PV3267) were purchased from Thermo Fisher Scientific. For kinase reaction, 200 ng of each CDK/Cyclin complex was incubated with 1 µg of His-PNKP in kinase reaction buffer (25 mM Tris-HCl at pH 7.5, 5 mM MgCl$_2$, 5 mM sodium pyrophosphate, 2 mM ATP, and 2 mM DTT) for 1 hr at 30°C. Reactions were quenched by adding 2× SDS sample buffer and boiling for 10 min at 98°C. Samples were processed for western blotting and detected using specific antibodies.

## Statistical analysis

Statistical analysis was performed using either GraphPad Prism 8 (GraphPad Software Inc) or Microsoft Excel. Unpaired (two-tailed) $t$-tests were used to analyze the statistical significance of differences between the two experimental groups. One-way ANOVA followed by post hoc tests were used to analyze the statistical significance between multiple experimental groups. Sample scales are indicated in the figure legends. All experiments were independently performed at least three times with similar results. In all experiments, no statistical significance (ns) is defined as $p > 0.05$, * denotes $0.01 < p \leqq 0.05$, ** denotes $0.005 < p \leqq 0.01$, *** denotes $0.001 < p \leqq 0.005$, and **** denotes $0.0005 < p \leqq 0.001$.

## Acknowledgements

Authors thank Mr. Isao Yoda at Co$^{60}$ radiation center, Drs. Kimitoshi Denda, Hiromi Yanagihara, Hirofumi Nakano, and Daisuke Morishita for technical assistance, and Matsumoto laboratory member for critical discussion. Authors also thank to Dr. Michael Weinfeld to provide purified PNKP proteins. This work was supported by The Uehara Memorial Foundation [to MS], Takeda Science Foundation [to MS], Kato Memorial Bioscience Foundation [to MS], Japan Atomic Energy Agency [to MS], and Chubu Electric Power [to MS], Tokyo Tech Academy for Co-creative Education of Environment and Energy Science [to KT], Tokyo Tech Academy for Leadership [to KT], Grant-in-Aid for Scientific Research from Japan Society for the Promotion of Science [Grant Numbers JP22K12369 to MS, JP15H02817, JP17K20042, JP20H04334 to YM and JP18K11642 to MI], Grant-in-Aid for Japan Society for the Promotion of Science Fellows [Grant Number JP20J13601 to KT], Japan Society for the Promotion of Science Overseas Research Fellowships [to KT] and Radiation Effects Association [to MI].

## Additional information

### Funding

| Funder | Grant reference number | Author |
|---|---|---|
| Tokyo Tech Academy for Co-creative Education of Environment and Energy Science | | Kaima Tsukada |
| Japan Society for the Promotion of Science | JP20J13601 | Kaima Tsukada |
| Kato Memorial Bioscience Foundation | | Mikio Shimada |
| Japan Atomic Energy Agency | | Mikio Shimada |
| Chubu Electric Power Company | | Mikio Shimada |
| Tokyo Tech Academy for Leadership | | Kaima Tsukada |

| Funder | Grant reference number | Author |
|---|---|---|
| Uehara Memorial Foundation | | Mikio Shimada |
| Takeda Science Foundation | | Mikio Shimada |
| Japan Society for the Promotion of Science | JP22K12369 | Mikio Shimada |
| Japan Society for the Promotion of Science | JP15H02817 | Yoshihisa Matsumoto |
| Japan Society for the Promotion of Science | JP17K20042 | Yoshihisa Matsumoto |
| Japan Society for the Promotion of Science | JP20H04334 | Yoshihisa Matsumoto |
| Japan Society for the Promotion of Science | JP18K11642 | Masamichi Ishiai |
| Radiation Effects Association | | Masamichi Ishiai |

The funders had no role in study design, data collection, and interpretation, or the decision to submit the work for publication.

### Author contributions

Kaima Tsukada, Conceptualization, Data curation, Software, Formal analysis, Supervision, Validation, Investigation, Visualization, Methodology, Writing – original draft, Project administration, Writing – review and editing; Rikiya Imamura, Conceptualization, Data curation, Formal analysis, Investigation, Visualization, Methodology, Writing – review and editing; Tomoko Miyake, Data curation, Software, Formal analysis, Validation, Investigation, Visualization, Methodology, Writing – review and editing; Kotaro Saikawa, Mizuki Saito, Naoya Kase, Lingyan Fu, Investigation; Masamichi Ishiai, Yoshihisa Matsumoto, Funding acquisition, Investigation; Mikio Shimada, Conceptualization, Resources, Data curation, Software, Formal analysis, Supervision, Funding acquisition, Validation, Investigation, Visualization, Methodology, Writing – original draft, Project administration, Writing – review and editing

### Author ORCIDs

Kaima Tsukada ⓘ https://orcid.org/0000-0002-6725-9514
Tomoko Miyake ⓘ https://orcid.org/0000-0003-3389-4007
Masamichi Ishiai ⓘ https://orcid.org/0000-0003-4313-9945
Yoshihisa Matsumoto ⓘ https://orcid.org/0000-0002-0758-290X
Mikio Shimada ⓘ https://orcid.org/0000-0003-1980-9187

### Decision letter and Author response

Decision letter https://doi.org/10.7554/eLife.99217.sa1
Author response https://doi.org/10.7554/eLife.99217.sa2

---

## Additional files

### Supplementary files

Supplementary file 1. List of DNA primers. Sequences of DNA oligonucleotide primers for mutagenesis of PNKP. 'F' and 'R' indicate forward and reverse sequences, respectively.

Supplementary file 2. List of siRNAs. Oligonucleotide sequences of siRNAs for specific depletion with indicated proteins.

MDAR checklist

### Data availability

All data generated or analyzed during this study are included in the manuscript and supporting files; source data files have been provided for figures and figure supplements.

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
