## [Editor Report]

PNKP (polynucleotide kinase phosphatase) has dual functions of adding or removing phosphate groups at broken DNA ends for DNA damage repair, such as base damage or strand breaks. Here the authors provide convincing evidence that PNKP is also involved in the DNA replication process, particularly in processing Okazaki fragment ends. This is an important finding that expands our understanding of PNKP function, showing that it extends beyond DNA damage repair.

---

## [Decision Letter]

[Editors' note: this paper was reviewed by Review Commons.]

---

## [Author Response]

General Statements

We would like to thank all three reviewers for their positive comments on the value of our work and for their helpful suggestions. We believe that the planned revisions, based on the comments by the reviewers, will substantially enhance the quality of our manuscript. In summary, we are mainly planning to perform four experiments: iPOND assay using XRCC1-depleted cells, iPOND assay using PNKP-T118A expressing cells, DNA repair assay in PNKP-T118A expressing cells throughout cell cycle, and an alternative method of SSB repair assay using BrdU incorporation, as well as several control/follow-up experiments suggested by the reviewers.

Our detailed responses to all their comments are given in following sections.

Description of the revisions

Reviewer #1PNKP is one of critical end-processing enzymes for DNA damage repair, mainly base excision & single strand break repair, and double strand break repair to a certain extent. This protein has dual enzyme function: 3' phosphatase and 5' kinase to make DNA ends proper for ligation. It has been demonstrated that PTM of PNKP (e.g., S114, S126), particularly phosphorylation by either ATM or DNAPK, is important for PNKP function in DNA damage repair. The authors found a new phosphorylation site, T118, of PNKP which might be modified by CDK1 or 2 during S phase. This modification of phosphorylation is involved in maintenance and stability of the lagging strand, particularly Okazaki fragments. Loss of this phosphorylation could result in increased single strand gaps, accelerated speed of fork progression, and eventually genomic instability. And for this process, PNKP enzyme activity is not that important. And the authors concluded that PNKP T118 phosphorylation is important for lagging strand stability and DNA damage repair.Major comments1. In general, enzymes have protein interactions with its/their substrates. If PNKP is phosphorylated by either/both CDK1/2, the protein interaction between these would be expected. However, the authors did not provide any protein interactions in PNKP and CDKs.

Thank you for your suggestion. We have performed GFP-pulldown assays using cell extracts from HEK293 cells expressing GFP-WT-PNKP or GFP-T118A-PNKP and 3XFLAG-CDK1 or 3XFLAG-CDK2. We confirmed the interactions of PNKP and CDK1/2 by western blotting with FLAG antibody (Figure 4—figure supplement E). T118A mutation on PNKP affects its interaction with CDK1/2. Further, the result may suggest that PNKP preferentially interacts with CDK2 over CDK1, although the baseline expression level of CDK2 were slightly higher than that of CDK1.

2. It is not clear how T118 phosphorylation is involved in DNA damage repair itself as the authors suggested. The data presenting the involvement of T118 phosphorylation in this mechanism are limited. This claim opens more questions than answers. CDK1/2 still phosphorylates T118 in this DNA damage repair process? What would happen to DNA damage repair in which PNKP involves outside of S phase in terms of T118 phosphorylation?

Thank you for your comment. We agree with your point. To prevent for dispersion of the focus of this manuscript, we re-considered and re-structured the total layout of the revised manuscript (as suggested by reviewer 3). You can find that we now mainly focused on the role of PNKP in DNA replication, which is an unexplored role of PNKP. We believe that the revised version of the manuscript is more straightforward for explaining an important role of PNKP in processing of Okazaki fragments during DNA replication.

Briefly, the phosphorylation of PNKP of T118 is CDK1/2-dependent process, meaning that the phosphorylation peaks at S phase, which is mainly shown in new figure 4 and figure 4—figure supplement. Therefore, it does not seem a DNA damage responsive phosphorylation. Along with general DNA repair and genomic instability assays in T118A expressing PNKP KO cells (figure 5—figure supplement C-F), we have performed H2OS-induced SSB assay using EdU (S phase) pulse labelling in PNKP KO cells expressing PNKP T118A mutant, then we measured the ADP-ribose intensity in EdU negative cells (outside of S phase as suggested by the reviewer) in new figure 5—figure supplement E (also below). Interestingly, we see that T118A cells behave like PNKP KO cells in S phase population, but not outside the S phase, indicating that T118 phosphorylation mainly involves in DNA repair process against H_2_O_2_-induced exogenous DNA damage in S phase. Future study will be needed to test this regulation of DNA repair processes via the PNKP phosphorylation.

3. Along the same line with #1/2 comments, the recruitment of PNKP to the damage sites is XRCC1 dependent. Is not clear whether PNKP recruitment to gaps on the lagging strand is XRCC1 independent or dependent. It might be interesting to examine (OPTIONAL)

Thank you for an important suggestion. XRCC1 acts as a scaffold of PNKP and is required for recruitment of PNKP for canonical SSB repair, although we propose that PNKP is involved in two pathways in DNA replication: PARP1-XRCC1-dependent ssDNA gap filling pathway and canonical Okazaki fragment maturation pathway working with FEN1. It is still important to address how XRCC1 is required for PNKP recruitment to the single-strand gaps on nascent DNA. Therefore, we have added iPOND analysis in XRCC1 knock down + GFP-WT-PNKP expressed HEK293 cells in Figure 4F. The result shows that XRCC1-depletion reduces an amount of PNKP on nascent DNA, although this reduction seems lesser extent than XRCC1 reduction by siRNA. This may indicate that PNKP has a XRCC1-independent alternative recruitment pathway during DNA replication.

Minor comments1. In results: 'Generation of PNKP knock out U2OS cell line'- In figure S2A; There are no data regarding diminishing the phosphorylation of g-H2AX.

Thank you for your suggestion. We have added pH2AX blot data in figure 1—figure supplement 2A (all reviewers requested).

- By showing data in figure S2B/C/D/E, the authors describe 'PNKP KO cells impaired the SSBs repair activity'. However, as the authors mentioned in this manuscript, PNKP could bind to either XRCC1 or XRCC4. Also for this experiment, IR had been applied, which induces DNA double strand breaks. Therefore, it is not certain that the authors' description is fully supported by these data presented. Perhaps, SSB inducing reagents should be used instead of IR.

In the figure S2B/C/D/E, we used γ-ray as IR source, which classified as low energy transfer irradiation, which mainly act as indirect effect to the DNA. It is estimated γ-ray induce DNA damage as 60-80% SSBs and 20-40 % DSBs. We believe that our results are reasonable. In addition to these mentioned above, we have performed native BrdU incorporation assay with H_2_O_2_ treatment to more specifically assess SSBs repair activity. In line with our data used IR, PNKP KO cells showed a markedly higher BrdU incorporation level upon H_2_O_2_ treatment compared to that in PNKP WT cells (new Figure 1—figure supplement 2D-E). Therefore, we suppose that our updated description, “We then assessed~~~, indicating that the loss of PNKP leads to reduced ability of SSB repair in cells (Figure 1—figure supplement 2C-F)”, is now strongly supported by this result. As a result of re-structuring/cutting down of figures (as suggested by the reviewer 2 and 3), previous figure S2D/E were removed from the panel.

- Is there any FACS analysis data to support the description of the last sentence 'especially the phosphorylation of PNKP T118, is required for S phase progression and proper cell proliferation'?

Thank you for your suggestion. We have performed the FACS analysis data of cell cycle profiles in PNKP KO cells expressing GFP, GFP-PNKP WT or T118A. Unfortunately, transient transfection of the plasmids affects their cell cycle. We were not able to get further supporting evidence of the sentence you mentioned. Therefore, we have modified the sentence to “especially the phosphorylation of PNKP T118, is required for proper cell proliferation” to make our point clearer.

**Author response image 1. sa2fig1:** 

- Is there any difference (except for PARGi exposure time?!) between figure S2B/C and S2D/E? Both data show increased ADP ribose after IR. It seems redundancy. Also it is hard to imagine that there is absolutely no sign of ADP ribose after IR w/o PARGi treatment (figure S2D).

The previous figure S2B/C show spontaneous single strand DNA breaks (SSBs) in PNKP KO cells, on the other hand, the previous figure S2D/E show ectopic SSBs induced by IR exposure in PNKP KO cells. Poly-ADP ribosylations are immediately removed from SSB sites after repair as demonstrated previously (Tsukada, et al., PLoS One 2019, Kalasova et al., Nucleic Acids Research, 2020), although not zero (low level), it is very difficult to detect without PARGi treatment. However, during this revision, we have investigated this by an alternative experimental method, a BrdU incorporation assay (in new figure 1—figure supplement 2D-E), which is a more direct method to detect single-stranded DNA. In line with the previous figure 2D/E, new figures show that PNKP KO cells exhibit impaired DNA repair ability against exogenous DNA damage induced by H_2_O_2_, a SSB inducer. Therefore, we replaced the previous figure S2D/E to new figure S2D-E due to redundancies between those results. We believe these new data help for readers to understand the effect of endogenous or exogenous damage in PNKP KO cells.

2. Legend for figure S3 – typo!

Thank you for your suggestion about typo. The legend for figure 3—figure supplement is corrected as “Protein expression of PNKP mutants in U2OS cells”.

- In figure S3A/B, it is quite interesting that the PNKP antibody used for this analysis can detect all truncated and alanine substituted PNKP proteins. It might be helpful to indicate for other researchers which antibody used (Novus; epitope – 57aa to 189 aa or Abcam; epitope not revealed).

In figure 3—figure supplement A/B, Novus PNKP antibody was used for all blots. We indicated this in the figure legend as “PNKP antibody (Novus: NBP1-87257) was used for comparing expression levels of endogenous and exogenous PNKP”.

3. In results: 'PNKP phosphorylation, especially of T118 ~~~ proliferation'- In the fork progression experiment (figure 2C), is there any statistical difference between D2 and D3/4 expressing cells?

Thank you for your suggestion. The corresponding figure (previous figure 2C) is now figure 3C. In the new figure 3C. We performed statistical analysis as the reviewer suggested. Statistical analysis shows that there are no significant differences between D2 and D3/D4. Meanwhile, there are significant differences between WT and D3(P<0.01), D4 (P<0.001), indicating that D3 and D4 mutants are not able to rescue this phenotype.

- What is the basis of the description 'Since the linker region of PNKP is considered to be involved in fork progression'? Any reference?

Thank you for your comment. This sentence was considered based on the results in figure 3A-C. However, we suppose this description may be overstated, we therefore modified the description to “D2 mutant-expressing cells showed slower proliferation than cells expressing WT PNKP and other mutants, although D3 and D4 exhibited mildly slower cell proliferation (WT vs D3: P=0.1737; WT vs D4: P=0.4523). Furthermore, D3, D4 as well as D2 mutant-expressing cells showed increased tract lengths compared to WT and D1 mutant-expressing cells, indicating that in addition to the enzymatic activity of PNKP, the linker region also plays a crucial role in proper fork progression.”.

- In figure 3B: pS114-PNKP (also pS15-p53) is DNA damage inducible. In this experiment, was DNA damage introduced? Roscovitine could hinder DNA repair process, but not inducing DNA damage itself.

Thank you for your suggestion. DNA damage induction was not applied in this experiment. We agree that this panel makes confusing. We think that endogenously S114-PNKP (also S15-p53) might be phosphorylated slightly but not significant, although this is not the scope of this manuscript. This result showing that phosphorylated-T118 is reduced by Roscovitine treatment maybe redundant as we also have a result of in vitro phosphorylation assay using several combinations of CDKs and Cyclin proteins, which is a cleaner experiment to prove which CDK/Cyclin complex is directly controlling the T118 phosphorylation (in new figure 4B). Furthermore, we performed a GFP-PNKP pulldown assay to investigate the interaction between PNKP and CDK1/2 in cells (in new figure S4E). In order to answer reviewer 3’s comment suggesting us to remove redundant figures from the panel, we removed this WB blot result from the panel due to the redundancy of these data.

4. In results: 'CDKs phosphorylate T118 of PNKP ~~~ replication forks'- In figure 3A, Is there any change in total PNKP (both GFP-tagged & endogenous) level?

Thank you for your suggestion. We agree with your comment. We have added the both GFP-PNKP and endogenous PNKP expression analysis in different cell cycle population in asynchronized and synchronized cells (G1, S, G2/M samples). As you can see in the new figure 4A, GFP-PNKP expression levels are consistent throughout cell cycle. Therefore, we believe that T118 phosphorylation is peaked in S phase. However, endogenous PNKP expression levels were slightly changed through cell cycle, peaking at S phase, which indicate that endogenous PNKP expression level or its stability may be regulated in a cell cycle-dependent manner. This is an interesting observation, but it is out of scope in this study, so follow-up study will be required for investigating the regulatory mechanism of PNKP protein stability.

5. In results: ‘Phosphorylation of PNKP at T118 ~~~ between Okazaki fragments’- In figure 4D, What happens in the ADP-ribose level, when T118D PNKP is expressed?

Thank you for your suggestion. This is an interesting question. We performed ADP-ribosylation assay in PNKP KO cells expressing PNKP WT and T118D (new figure 5—figure supplement A/B). The phospho-mimetic mutant T118D-expressing PNKP KO cells behave like the PNKP KO cells expressing PNKP WT, which indicates the importance of the phosphorylation of PNKP on T118 in dealing with endogenous and FEN1i-induced single-strand DNA gaps.

6. In results: 'Phosphatase activity of PNKP is ~~~ of Okazaki fragments'- In figure 5C, any statistical analysis between WT-PNKP KO vs D171A-PNKP KO or K378A-PNKP KO has been done?

Thank you for your comment. Statistical analysis shows P<0.1 between WT PNKP vs D171A and no statistical significance between WT PNKP vs K378A PNKP in IdU tract length of these samples, although p-value between PNKP KO vs WT PNKP, D171A, or K378A are P<0.0001, ns, and P<0.1, respectively. Those data indicates that D171A and K378A mutants are not fully capable to compliment ability of fork progression comparing to PNKP WT cells.

7. In results: 'PNKP is involved in post replicative single-strand DNA gap-filling pathway'- The description regarding data presented in figure 6 is not clear enough. These data might suggest that wildtype U2OS does not have SSB which is a substrate for S1 nuclease (except under FEN1i and PARPi treatment), whereas PNKP KO has SSB during both IdU and CIdU incorporation, so that S1 nuclease treatment dramatically reduces the speed of fork formation in PNKP KO cells. Also In figure 6B/C/D, adding an experimental group of PNKP KO with S1 nuclease + PARPi might help to understand the role of PNKP during replication better. Also these additional data could support the description in discussion 'Furthermore, PNKP is required for the PARP1-dependent single-strand gap-filling pathway ~~~ DNA gap structure'.

Thank you for your suggestion. We agree with reviewer’s comment and suggestion. Since this point is also raised by reviewer 3, we added the rationale of the experiment (new figure 2C) and more detailed description about the results. We have also revised our representation in text followed by the comment. In addition to revising the text, we have added experiment groups of PNKP KO with S1 nuclease with/without PARPi treatment as the reviewer suggested (new figure 2D). PNKP KO cells showed significantly smaller IdU/CldU ratio upon S1 nuclease treatment regardless of the presence of PARP1 inhibitor (the sixth and eighth lane from the left). Although PARP1 inhibitor treatment in U2OS cells creates substrates for S1 nuclease (the third and fourth lane from the left), the extent of post-replicative single-strand DNA gaps in PNKP KO cells were further increased (the fourth and eighth lane from the left). This result indicates that PNKP is involved in the single-strand DNA gap filling pathway.

8. In results: 'Phosphorylation of PNKP at T118 is essential for genome stability'- In figure S8C, Did you measure g-H2AX foci disappearance for later time point, such as 24 hrs after DNA damage? Is not clear whether non-phosphorylated PNKP at T118 inhibit DNA damage repair or make it slower? How does T114A-PNKP behave in this experimental condition? T114 is well known target of ATM/DNAPK for DDR & DSB repair.

Thank you for your suggestion. We agree with your point. It is very important to analyze whether T118A mutant shows delayed or total loss of DSB repair ability. We added the measurement of pH2AX foci at 24 hrs after IR in PNKP KO cells expressing GFP, WT-PNKP, T118A-PNKP. Although the analysis of pS114 PNKP is previously reported (Segal-Raz et al., EMBO reports, 2011 and Zolner et al., Nucleic Acids Research, 2011), we also performed pH2AX assay in PNKP KO cells expressing S114A-PNKP as a control (figure 5—figure supplement F). The result shows that cells expressing S114A or T118A mutant still have increased gH2AX foci levels at 24 hrs after IR exposure, indicating that these mutants are delayed in repair of IR-induced DNA damage at least up to 24 hrs.

9. The result shown in figure S9 should be described in the result section, not in the Discussion section.

Thank you for your suggestion. This is a point also raised by Reviewer 3. Since we re-considered the layout of the manuscript, we now removed this figure out from the manuscript (as reviewer 3 suggested).

10. In discussion, 'In contrast, the T118A mutants showed the absence of both SSBs and DSBs repair (Figure S7) : figure S7 does not indicate what the authors describe.

Thank you for pointing out this error. This should have referred to figure S8 instead of figure S7 in the previous version. Through the reordering of figures during this revision, the corresponding figure became new figure 5—figure supplement C-F.

11. In addition, the same sentence in discussion: No evidence demonstrate that 'the absence of both SSBs and DSBs repair', and the following sentence is not clear.

This is same point with above. We have corrected this mis-referencing and revised the sentence to “The T118A mutant demonstrated reduced repair abilities of both exogenous SSBs and DSBs (Figure 5—figure supplement C-F). However, T118A-expressing cells only show increased ADP-ribose intensity in the S phase population upon H_2_O_2_ treatment. Future studies will be able to elucidate the function of T118 phosphorylation in response to exogenous DNA damage.” for better explanation of the result.

12. In discussion, 'Because both CDK1/cyclin A2 and CDK2/cyclin A2 are involved in PNKP phosphorylation, cyclin A2 is likely important for these activities': It is not clear what this description intends? Is 'cyclin A2' important in what stance?

This description is coming from observations in figure 4A-C. Since both CDK1 and CDK2 activities are cyclin A2 dependent, we speculated cyclin A2 is important for CDK1/CDK2 dependent PNKP T118 phosphorylation. We revised the description to “Since the activities of CDK1/Cyclin A2 and CDK2/Cyclin A2 complexes peak in the late S/G2 phase and S phase, respectively, we speculated that PNKP T118 is phosphorylated from the S phase to the G2 phase in CDK1/Cyclin A2- and CDK2/Cyclin A2-dependent manner (Figure 4B and C).”.

13. In discussion, 'This may be explained by the fact that mutations in the phosphorylated residue in the linker region are embryonic lethal': any reference to support this embryonic lethality?

Thank you for your suggestion. We agree with that this sentence is overwriting. We revise the sentence to “This observation may indicate that mutations in the phosphorylated residue (T118) in the linker region are potentially embryonic lethal due to the importance of T118 in DNA replication, which is revealed in the present study.”.

Referees cross-commentingI could see a similar degree of positive tendency toward the manuscript. I agree with the comments and suggestions in additional experiments made by reviewers 2 and 3. Those suggestions will improve an impact of the manuscript in the DNA damage repair field.Reviewer #1 (Significance (Required)):SignificanceThe authors discovered new phosphorylation site (T118) of PNKP which is an important DNA repair protein. This modification seems to play a role in maintenance of the lagging strand stability in S phase. This discovery is something positive in DNA repair field to expand the canonical and non-canonical functions of DNA repair factors.The data presented to support PNKP functions and T118 phosphorylation in S phase seem solid in general, yet it is not sure how much PNKP is critical in the Okazaki fragment maturation process which is known that several end processing enzymes (like FEN1, EXO1, DNA2 etc. which leave clean DNA ends.) are involved.These findings might draw good attention from researchers interested broadly in cell cycle, DNA damage repair, replication, and possibly new tumor treatment.My field and research interest: DNA damage response (including cell cycle arrest and programmed cell death), DNA damage repair (including BER, SSBR, DSBR)

Thank you very much for your positive comment. As you mentioned, there are several other end processing enzymes that seem to be involved in canonical Okazaki fragment maturation pathway, however, none of those enzymes is reported as a protein involved in the gap-filling pathway as well. Therefore, the role(s) of PNKP in DNA replication are very outstanding as PNKP could be involved in two separate pathways, the canonical Okazaki fragment maturation and a back-up gap-filling repair process. As you suggested, we have added several experiments such as iPOND experiments using XRCC1-depleted cells, analysis of DNA repair ability of PNKP T118A mutant throughout cell cycle and S1 nuclease DNA fiber assays in PNKP KO cells with/without PARP inhibitor treatment, to reveal how much PNKP is critical in the Okazaki fragment maturation. We believe that those experiments make the conclusion and this manuscript more solid and convincing.

Reviewer #2 (Evidence, reproducibility and clarity (Required)):Polynucleotide kinase phosphatase (PNPK) participates in multiple DNA repair processes, where it acts on DNA breaks to generate 5'-phosphate and 3'-OH ends, facilitating the downstream activities of DNA ligases or polymerases.This manuscript identifies a CDK-dependent phosphorylation site on threonine 118 in PNKP's linker region. The authors provide some convincing evidence that this modification is important to direct the activity of PNPK towards ssDNA gaps between Okazaki fragments during DNA replication. The authors monitored protein expression levels, enzymatic activity, the growth rate and replication fork speed, as well as the presence of ssDNA damage to make a comprehensive overview of the features of PNKP necessary for its function.Overall, the conclusions are sufficiently supported by the results and this manuscript is relevant and of general interest to the DNA repair and genome stability fields. Some level of revision to the experimental data and text would help strengthen its message and conclusions.Major points:1. In an iPOND experiment the authors detect the wt PNKP and the T118 phosphorylated form at the forks and conclude that this phosphorylation promotes interaction with nascent DNA (Figure 3E). An informative sample to include here would have been the T118A mutant. Based on the model proposed, the prediction would be that it would not be associated with the forks, or at least, associated at reduced levels compared to the wt.

Thank you for your suggestion. We agree with your comment. We have added the iPOND analysis in PNKP T118A expressing PNKP KO cells to confirm that pT118 is important for recruitment of PNKP at nascent DNA. As you can see in the figure 4E, the T118A mutant shows a reduced recruitment of PNKP at nascent DNA, indicating phosphorylation on T118 is important for the recruitment.

2. The quality of the gels showing the phosphatase and kinase assays in Figure 5 could be improved to facilitate quantification of the results. The gel showing the phosphatase activity has a deformed band corresponding to K378A mutant. The gel showing the kinase activity seems to be hitting the detection limits, and the overall high background might influence the quantification of D171A mutant in the area of interest. The authors should provide a better quality of these gels, focusing on better separation (running them longer, eventually with a slightly increased electric current) and higher signal of the analyzed bands (longer incubation phosphatase/kinase prior to quenching or loading higher amount of DNA).

Thank you for your suggestion. We agree with your suggestion. This phosphatase and kinase assay could be improved. We have performed this assay again followed by reviewer’s suggestions. New gels are shown in figure 6—figure supplement C.

3. The authors sometimes make statements like: "a slight increase, slightly increased, relatively high" without an evaluation of the statistical significance for the presented data. An example of such a statement is: "T118A mutant-expressing cells exhibited a marked delay in cell growth, which was not observed for S114A, although T122A, S126A, and S143A were slightly delayed," based on the figure 2E. A similar comment applies also to figures 4A, 5A, 5E. Whenever possible, the authors should include also an evaluation of the statistical significance in the statement.

Thank you for your suggestion. According to reviewer’s suggestion, we checked manuscript, revised representation and added the evaluation of statistical significance in the statement where it’s applied.

Minor revisions:4. I could not find a gH2AX blot for figure S2A.

Thank you for your suggestion. We have added pH2AX blot data in figure 1—figure supplement 2A.

5. Sometimes there are incorrect references to the figures in the discussion (e.g. FigS9A, B, and C, are called out instead of E, F and G), a similar issue is found 4 lines below in the same page.

Thank you for pointing out these errors. We checked the references in the discussion and corrected to the appropriate references.

6. The authors established two PNKP-/- clones and supported it with sequencing and several functional observations. However, the C-terminal antibody appears to detect lower-intensity bands (Figure 1A). Can authors comment on those bands?

Thank you for your comment. One possibility of this band is non-specifically recognized bands. To improve this problem, we tried to run electrophoresis for longer time to separate this band. The improved blot is now shown in new Figure 1A.

7. Based on the data in Figure 3A the authors suggest that pT118-PNKP follows Cyclin A2 levels, but this does not appear very clearly in the gel, especially for the last point. Even though the results are convincing, the authors should rephrase the conclusions of Figure 3A to reflect better the results.

Thank you for your suggestion. We agree that this phrase is overwriting. We revised the conclusion to “pT118-PNKP was detected in asynchronized cells but increased particularly in the S phase, similar to Cyclin A2 expression levels. However, the reduction of pT118, possibly due to dephosphorylation of T118, was not as robust as the reduction in Cyclin A2 expression levels at the 12-hour time point. This effect was very weak during mitosis, suggesting that T118 phosphorylation plays a specific role in the S phase.”.

8. Why the S1 nuclease data on DNA fibers do not show the same level of epistasis with the Fen1i, as do those on ADP-ribosylation?

Firstly, because FEN1 dependent canocical Okazaki fragment maturation and PARP1-XRCC1 dependent gap-filling pathway are different pathways (Vatsiankova et al., 2022), FEN1i and PARPi treatment resulted in an additive effect in S1 nuclease data in PNKP WT cells in figure 2C. Our results suggest that PNKP is involved in both pathways mentioned above. Secondly, this could be due to the nature of these experimental methods. In new figure 2C, we used S1 nuclease to digest post-replicative single-strand DNA gaps to investigate length of single-strand DNA gaps at the single DNA fiber level, whereas the ADP-ribosylation assay is a method to detect general ADP-ribosylation levels in each cell. Therefore, those results may show length of single-strand DNA gaps and numbers of single-strand DNA gaps, respectively, which are not fully same phenomenon. To facilitate better understanding, we added graphical scheme in new figure 2E (a similar problem was raised by Reviewer 3 below) and revised the description of the result.

9. I did not find a reference to what seems to be a relevant work in this topic: PMID: 22171004

Thank you for pointing our this error. We have added the ref (Coquelle et al., PNAS, 2011) in Introduction section.

Referees cross-commentingI agree with all the comments from the reviewers 1 and 3.Reviewer #2 (Significance (Required)):Significance:The manuscript identifies a CDK phosphorylation site in a relevant DNA repair protein. The experiments on this part are elegant and convincing. It seems that this phosphorylation is important during DNA replication and there is some supporting evidence in this point, although not as robust, meaning that it is not clear whether this phosphorylation is controlling specifically the recruitment to Okazaki fragments, or a general role in DNA repair. Maybe if they see a reduced recruitment of the T118A mutant to the forks (iPOND experiment) this would further increase the impact.This work will be relevant to the basic research, especially in the fields of DNA repair and DNA replication.My expertise: DNA replication, genome stability, telomere biology.

Thank you very much for your positive comment. As you suggested, we performed a numbers of experiments mentioned above including an iPOND assay using PNKP T118A mutant, which shows reduced recruitment of the T118A mutant to the replication forks. We believe that results of these experiments pin down whether the phosphorylation of PNKP on T118 is controlling its recruitment to Okazaki fragments specifically or single-strand DNA gaps in general, and solidify the conclusion of the manuscript.

Reviewer #3 (Evidence, reproducibility and clarity (Required)):Tsukada and colleagues studied the role of PNKP phosphorylation in processing single-strand DNA gaps and its link to fork progression and processing of Okazaki fragments.They generated two PNKP KO human clonal cell lines and described defects in cell growth, accumulation in S-phase, and faster fork progression. With some elegant experiments, they complement the KO cell lines with deletion and point mutants for PNKP, identifying a critical phosphorylation site (T118) in the linker regions, which is important for cell growth and DNA replication.They show that phosphorylation of PNKP peaks in the mid-S phase. CDK1 and CDK2/ with Cyclin A2 are the two main CDK complexes responsible for this modification. With the IPOND experiment, the author shows that PNKP is recruited at nascent DNA during replication.They described increased parylation activity in PNKP KO cells, and by using HU and emetin, they concluded that this increased activity depends on replication and synthesis of Okazaki fragments.Interfering with Okazaki fragment maturation by FEN1 inhibition is epistatic with PNKP KO (and T118A) in influencing parylation activity in the S phase and fork progression. The authors try to understand by mutant complementation which of the two functions (Phosphatase vs Kinase) is important in processing OF, and they propose a primary role for the phosphatase activity of PNKP. They also show that T118 is important in controlling genome stability following different genotoxic stress. Finally, by coupling the measurement of fork progression with PARP/FEN1 inhibitors and S1 treatment, they propose a role of PNKP in the post-replicative repair of single-strand gaps due to unligated OF.Here are my major points:– The authors use a poly ADP ribose deposition measurement to estimate SSB nick/gap formation. Even if PARP activity is strictly linked to SSB repair, ADP ribosylation does not directly estimate SSB/nick gap formation. In addition, in FiguresS2A, B, and C, the authors use IR and PARG inhibition to measure poly-ADP ribosylation in WT and PNKP KO cells. IR produces both SSB and DSB. A better and cleaner experiment would be to directly measure SSB formation (with alkaline comet assay, for example) in combination with treatments that are known to mainly cause SSB (H_2_O_2_, or low doses of bleomycin).

Thank you for your suggestion. Previous our report published in EMBO Journal (Shimada et al., 2015), we showed SSBs and DSBs repair defect in PNKP KO MEF with comet assay (both alkaline and neutral) after IR and H_2_O_2_ treatment. In addition to those observations, to measure SSB formation more directly, we performed BrdU incorporation assay in PNKP WT and KO cells treated with H_2_O_2_ (figure 1—figure supplement 2D-E), showing that PNKP KO cells exhibits significantly higher BrdU incorporation upon H_2_O_2_ treatment compared to that in PNKP WT cells. BrdU staining under an undenatured condition has now been commonly used and is a more direct method to detect ssDNA nick/gap formation. We believe that the importance of PNKP in SSB repair is sufficiently supported by all data such as previous comet assays in PNKP KO MEF cells and the SSB repair assay in human cells using BrdU incorporation.

– The manuscript would benefit from substantially restructuring the figures' order and panels. Before starting the T118 part, the authors could create several figures to explain the main consequences of the loss of PNKP. A figure could be focused on DSB-driven genome instability (Figure 1 + Figure S8 and S9). Then, a figure for the single-strand break and link to the S-phase. For example, by using data from Figure 6 and showing only WT vs PNKP KO +\- Nuclease S1 (without FEN1 or PARP inhibitors), the authors could easily convince the readers that loss of PNKP leads to the accumulation of single-strand gaps. Only in the second part of the manuscript could they introduce all the T118 parts.

Thank you for your suggestion. The layout of the manuscript makes reviewers feeling confusing. As you can find in our revised manuscript, we reconsidered the total layout of the manuscript carefully. Briefly, as you suggested, we made a figure explaining general consequences of the loss of PNKP (figure 1—figure supplement 2), then a figure (new figure 1) showing that PNKP-deficient cells exhibit an accumulation of single-strand DNA gaps in S-phase, followed by a figure (new figure 2) showing that the single-strand DNA gaps in PNKP-deficient cells are caused by problems during Okazaki-fragment maturation process. Then, finally we introduce all the T118 parts (new figure 3-5) and the PNKP’s enzymatic activity part (new figure 6).

– I understand the use of a FEN1 inhibitor to link the PNKP KO phenotype to OF processing, but this drug does not either rescue or exacerbate any of the phenotypes described by the authors. It seems to have just an epistatic effect everywhere. So, what other conclusion can we have if not that PNKO has a similar effect to FEN1? I think that the presence of this inhibitor in many plots complicates the digestion of several figures a little bit. Maybe clustering the data in a different way (DMSO on one side FEN1i on the other) would help.

Thank you for your suggestion. We agree that this data set is complicated. To facilitate better understanding, we changed organization of the data (new figure 5A, 5B, 5D/E, 6A and 6B/C) according to your suggestion and added graphical schemes in new figure 1, 2 and 6.

In terms of the other conclusion we can have from those experiments, the other conclusion is that PNKP may play two important roles in DNA replication: canonical Okazaki fragment maturation, which seems an epistatic effect with FEN1, and PARP1-XRCC1 dependent single-strand DNA gap filling pathway, which is required for repairing single-strand gaps between Okazaki fragments when canonical Okazaki fragment maturation pathway does not work properly (e.g., loss of FEN1 or PNKP). In new figure 2C, we show that a double treatment of FEN1i and PARPi in PNKP WT cells with S1 nuclease treatment shows extensive amount of digested DNA fibers, although a single treatment of either FEN1i or PARPi in PNKP WT cells with S1 nuclease treatment leads to only limited amount of digested DNA fibers, which indicates that two pathways regulated by FEN1 or PARP are coordinately required for preventing eruption of ssDNA gaps in DNA replication, consistent with previous study (Vaitsiankova et al. NSMB. 2022). On the other hand, PNKP KO cells with S1 nuclease treatment cause extensive amount of digested DNA fibers even without FEN1i and PARP1i treatments, also it is not further increased by FEN1i and PARPi treatment. Those results indicate that PNKP itself is involved in two pathways mentioned above. Therefore, loss of PNKP has a similar phenotype with loss of FEN1 in terms of canonical Okazaki fragment maturation, but also there is an additional effect in repairing those ssDNA gaps, which is created in FEN1 loss condition, but FEN1 seems not dealing with it.

– Figure S9 should be removed from the discussion. Additionally, the authors should consider whether they want to keep that piece of data in a manuscript that is already pretty dense. Why should we focus on additional linker residues and micro irradiation data at the end of this manuscript?

Thank you for your suggestion. We agree with your point and this is a point also raised by Reviewer 1. Therefore, we removed figure S9 out from the revised manuscript.

– I suggest using a free AI writing assistant. I think this manuscript would substantially benefit from one. As a non-native English speaker, I personally use one of them and find it extremely useful.

Thank you for your suggestion. Our manuscript was revised by a native speaker from an English correction company. However, for the revised manuscript, we discussed with native speakers as well as used a free AI writing assistant to improve the quality of the manuscript.

– The authors should consider and discuss the potential role of PNKP KO outside of the S-phase. In Figure 4C, while it is clear that poly ADP ribosylation is higher in S-phase, the effects of PNKP KO and complementation by WT or T118A are equally present. This would be more immediate if comparison, fold change, and statistical significance calculation were done within the same cell cycle phase instead of between cell stages. This is also clear by IF in Figure 4B. How do the authors explain this?

Thank you for your suggestion. We agree with reviewer’s suggestion. We compared intensities of ADP-ribose between cell lines in same cell cycle rather than between different cell cycles in a same cell line and added the respective fold-changes in the statement of figure 5C. Also, we agree with that poly ADP-ribose intensity is also changed outside of S phase between WT and T118A PNKP expressing PNKP KO cells. These results might reflect of PNKP function outside of S phase. Therefore, we have added the sentence “Of note, PNKP^−/−^ cells and PNKP T118A cells also showed higher ADP-ribose intensity outside the S phase (1.47-fold increase vs. WT-expressing cells), indicating that PNKP and T118 may play a role in preventing SSBs formation outside the S phase. Since FEN1 has been reported to function in R-loop processing, PNKP could also be involved in this process. Future studies of a role of PNKP in different cell cycle will be able to address this question.” to discuss about the function of PNKP outside the S phase. We have added the ref (Cristini et al., Cell Reports, 2019, and Laverde et al., Genes, 2022).

– In connection with the previous point, can the author provide the same quantification in Figure 4E also for G2/M and not only the S phase? This should give an estimate of the activity of FEN1 outside the S-phase. This is important because FEN1 has other functions apart from OF maturation, such as R loop processing (Cristini 2019; Laverde 2023)

Thank you for your suggestion. Here attached is the data of ADP-ribose intensity in cells outside the S phase as you suggested. Although the difference between with/without FEN1i treatment is much smaller than that in S phase, FEN1i treatment still induces increased ADP-ribose intensity in outside the S phase in PNKP WT-expressing PNKP^−/−^ cells as well, indicating that FEN1 has other functions outside the S phase. This finding is very interesting. However, the function of FEN1 in outside the S phase is outside the scope of this manuscript. Therefore, we would like to not put this data in the manuscript to avoid complicating the conclusion (as reviewer 3 also suggested).

– Why does FEN1 inhibition induce a faster fork progression in Figure 4 but not in Figure 5 and Fig6?

Yes, it does in previous figure 4 and figure 5 (new figure 2B, 5A and 6A). In PNKP WT cells, FEN1i-treated fibers (CldU) show an increased speed of forks compared to non-treated fibers (IdU). However, loss of PNKP and T118 phosphorylation themselves cause a faster fork progression even without FEN1i treatment, therefore the difference of speeds of forks before/after FEN1i treatment in PNKP KO and T118A cells disappears as both fibers grow faster than intact fibers in normal cells. In regard to figure 6 (new figure 2C), as you mentioned in a latter comment about this figure, those DNA fibers are potentially digested by S1 nuclease. Therefore, this is an experiment measuring the presence of single-strand DNA gaps on forks, but not directly measuring speeds of forks. Even so, DNA fibers from FEN1i-treated cells (CldU) with S1 nuclease shows similar length with fibers from untreated cells with S1 nuclease, whereas FEN1 inhibitor treatment accelerates a speed of forks in general (figure 4 and figure 5, assays without S1 nuclease), indicating that FEN1i treatment induces remaining of some ssDNA nicks/gaps which are substrates of S1 nuclease.

– How do the authors explain the impaired DNA gap binding activity of the phospho-mimetic T118D?

Thank you for your suggestion. We think that the appropriate timing of phosphorylation of PNKP T118 is important, while the phosphor-mimetic mutant T118D mimics consecutively phosphorylated situation that may result in incomplete complementation of PNKP function.

– I would like to see a representative fiber image from Figure 6. Additionally, in Figure 6, the author should not label the y-axis as CldU-fork speed. Nuclease S1 treatment destroys single-strand gaps (in vitro) and does not affect the fork speed (in vivo)

Thank you for your suggestion. We have added a representative fiber image in figure 1F. We also agree with that CldU fork speed is not a right label of y-axis as CldU fibers are potentially digested by S1 nuclease. We changed the y-axis label to “Normalized CldU tract length” in new figure 1G.

– Figure 5E: both mutants (kinase vs phosphatase) increase polyADP ribose intensity, while the title of this figure only emphasizes the phosphatase activity.

We agree with your comment. We have changed this subtitle to “Enzymatic activities of PNKP is important for the end-processing of Okazaki fragments”.

Minor points:– In Figure S1A, the author refers to P-H2AX, but I do not see this marker in the western blot.

Thank you for your suggestion. We have added pH2AX blot data in figure 1—figure supplement 2A.

The authors refer to Hoch Nature 2017 when referring to polyADP ribose IF + PARG inhibition. Should they not refer to Hanzlikova Mol Cell 2018?

Thank you for your suggestion. We have added ref (Hanzlikova et al., Mol Cell 2018).

– Statistical analysis should be performed on the cell cycle profile in Figure 1B

We performed statistical analysis to check whether there are significant differences of S phase population between WT and PNKP KO cells. There were significant differences between WT vs PNKP KO C1 (P<0.0499) and C2 (P<0.0485). We have added the statistics on the cell cycle profile in the corresponding figure 1—figure supplement 2H.

– The authors should not refer to fork degradation or protection as a given fact without assessing it in these conditions.

Thank you for your suggestion. We assume that this comment refers to the result section of figure 1G. We have changed representation in the section according to the reviewer’s suggestion.

Referees cross-commentingI agree with all comments from reviewer 1 and 2.Reviewer #3 (Significance (Required)):This is an interesting paper with generally solid data and proper statistical analysis. The figures are pretty straightforward. Unfortunately, the manuscript is dry, and the reader needs help to follow the logical order and the rationale of the experiments proposed. This is also complicated by the enormous amount of data the authors have generated. The authors should improve their narrative, explaining better why they are performing the experiment and not simply referring to a previous citation. Reordering panels and figures would help in this regard. Overall, with some new experiments, tone-downs over strong claims and a better explanation of the rationale behind experiments the authors could create a fascinating paper.

Thank you very much for your positive comment about the data/analysis and the logic behind the experiments provided in the manuscript. We agree with that a manner and a structure of the manuscript could be improved by reordering figures, cutting down some redundant experiments, adding better explanation of the rationale behind experiments, and toning-down some claims. With rewriting the manuscript as stated above and performing several additional experiments suggested by the reviewers, we believe that the revised manuscript became more convincing and fascinating.